# Empirical Bayes Trend Filtering
# Through a Variational Inference Framework

**Dongyue Xie**                                                  *dyxie@uchicago.edu*
*Department of Statistics*
*University of Chicago*

**Reviewed on OpenReview:** *https://openreview.net/forum?id=AHTz2mTlKk*

### Abstract

This paper introduces a novel framework for Bayesian trend filtering using an empirical Bayes approach and a variational inference algorithm. Trend filtering is a nonparametric regression technique that has gained popularity for its simple formulation and local adaptability. Bayesian adaptations of trend filtering have been proposed as an alternative method, while they often rely on computationally intensive sampling-based methods for posterior inference. We propose an empirical Bayes trend filtering (EBTF) that leverages shrinkage priors, estimated through an empirical Bayes procedure by maximizing the marginal likelihood. To address the computational challenges posed by large datasets, we implement a variational inference algorithm for posterior computation, ensuring scalability and efficiency. Our framework is flexible, allowing the incorporation of various shrinkage priors, and optimizes the level of smoothness directly from the data. We also discuss alternative formulations of the EBTF model, along with their pros and cons. We demonstrate the performance of our EBTF method through comprehensive simulations and real-world data applications, highlighting its ability to maintain computational efficiency while providing accurate trend estimation.

## 1 Introduction

Nonparametric regression methods have been widely used in many statistical applications such as spatial statistics, time series analysis, and survival analysis (Dabrowska, 1987; Gelfand & Schliep, 2016; Moulines et al., 2007). When the relationship between a predictor and a response variable is nonlinear, nonparametric regression can effectively capture the true underlying relationship by fitting a curve to the predictors. Classical nonparametric methods such as smoothing splines, B-splines, and kernel methods can be recast as penalized linear regressions and they are straightforward to fit (Härdle, 1990). However, they are not adaptive in the sense that they cannot adjust to local changes in the curve.

Trend filtering is a relatively new method for nonparametric regression. It penalizes the differences of adjacent signals using an $l_1$ penalty, and the penalty requires parameter tuning via cross-validation. Trend filtering was initially introduced as splines with higher-order total variation regularization (Steidl et al., 2006), without being named trend filtering. Later, trend filtering was independently introduced by Kim et al. (2009) as a modified version of Hodrick-Prescott (H-P) filtering (Hodrick & Prescott, 1997), that changes the penalty from $l_2$ to $l_1$ norm. A more statistical and theoretical study of trend filtering is provided by Tibshirani (2014), which showed that trend filtering achieves the minimax rate over a smoothness class defined by bounded total variation, mainly due to its ability to choose basis adaptively from data.

One of the earliest Bayesian adaptations of trend filtering is from Roualdes (2015), and the author borrowed ideas from Bayesian lasso (Park & Casella, 2008). Other shrinkage priors can be placed on the differences of the signals. Examples are the spike-and-slab (George & McCulloch, 1993), normal-gamma (Brown & Griffin, 2010), generalized double-Pareto (Armagan et al., 2013), horseshoe prior (Carvalho et al., 2009), and scale mixture of normal distributions (Faulkner & Minin, 2018). A dynamic shrinkage process and the

corresponding Bayesian trend filtering based on a dynamic linear model are proposed by Kowal et al. (2019). All of the above Bayesian methods use a Gibbs sampler for posterior inference. When the sample size $n$ is large, they suffer from high computational cost and low speed.

In this paper, we propose a novel framework for Bayesian trend filtering that is fast, locally adaptive, and accurate. Our method incorporates a shrinkage prior on the differences of the signal and employs an empirical Bayes method to estimate the prior by maximizing the marginal likelihood. The posterior distribution is computed using a variational inference algorithm. We highlight the advantages of our method as follows: 1. A fast and stable empirical Bayes trend filtering (EBTF), applicable to large-scale datasets; 2. Flexible shrinkage priors that adapt to the best shrinkage operator, not limited to the $l_1$ penalty; 3. Learns the level of "penalty" from data by optimization; 4. Naturally extends to more complex settings, such as sparse signal, which will be studied in Section 5. We provide a `Python` implementation of the method, and all the code and analysis are available in the package `ebtfPy` on GitHub (`https://github.com/DongyueXie/ebtfPy`). All the detailed derivations in this paper are in the Appendix D.

*Notation:* Denote the flat prior or improper prior for $\theta$ as $\theta \sim C(\cdot)$, and its density function is $p(\theta) \propto c$ over the support of $\theta$. A vector is denoted in bold such as $\boldsymbol{\beta}$, and when we need its elements, the vector is denoted as $\boldsymbol{\beta} = (\beta_i : i = 1, 2, ..., n)$. A diagonal matrix with its diagonal elements is denoted as $W = \text{diag}(w_i : i = 1, 2, ..., n)$.

## 2 Empirical Bayes Trend Filtering

In this section, we first give a brief review of the trend filtering problem, then present our models and algorithms. For a given integer $k \in \mathbb{N}$, the $k$th order trend filtering finds

$$\hat{\boldsymbol{\beta}} = \underset{\boldsymbol{\beta} \in \mathbb{R}^n}{\arg\min} \frac{1}{2}||\boldsymbol{y} - \boldsymbol{\beta}||_2^2 + \lambda||D^{(k+1)}\boldsymbol{\beta}||_1,$$

where $D^{(k+1)} \in \mathbb{R}^{(n-k-1)\times n}$ is the discrete difference operator of order $k+1$. When $k = 0$, the estimated sequence is piecewise constant, and the difference matrix is

$$D = \begin{bmatrix} -1 & 1 & 0 & \ldots & 0 & 0 \\ 0 & -1 & 1 & \ldots & 0 & 0 \\ \vdots & & & & & \\ 0 & 0 & 0 & \ldots & -1 & 1 \end{bmatrix} \in \mathbb{R}^{(n-1)\times n}. \tag{1}$$

For $k \geq 1$, the difference matrix is defined as $D^{(k+1)} = D^{(1)}D^{(k)}$, where $D^{(1)}$ is the $(n-k-1) \times (n-k)$ version of equation 1. When $k = 1$, the estimated sequence is piecewise linear; and twice-differentiable when $k = 2$.

Existing algorithms for solving the trend filtering problem include primal-dual interior point (Kim et al., 2009), path algorithm (Tibshirani & Taylor, 2011) and alternating direction method of multipliers (ADMM) (Ramdas & Tibshirani, 2016). The parameter $\lambda$ controls the smoothness of the estimated sequence and is often selected using cross-validation. Cross-validation for trend filtering is typically not random as the folds are often fixed. For a detailed description, see the `R` function `cv.trendfilter` (Arnold & Tibshirani, 2016).

### 2.1 Model formulation

We formulate trend filtering model as a dynamic linear model (DLM) data-generating process. Throughout this paper we will focus on the first-order sequence model, and we shall see that the general order model formulation is straightforward by using the corresponding-order difference matrix. We consider the following Bayesian variants of first-order trend filtering,

$$\begin{aligned} y_i|\beta_i &\sim N(\beta_i, \sigma^2 s_i^2), \text{ for } i = 1, ..., n, \\ \beta_1 &\sim C(\cdot), \\ (\beta_{j+1} - \beta_j) &\sim g(\cdot), \text{ for } j = 1, ..., n-1, \end{aligned} \tag{2}$$

where $C(\cdot)$ denotes the uniform distribution over the entire real line, $g(\cdot)$ is the prior on the difference between two consecutive means, $\sigma^2$ is the unknown random error variance and $s_i^2$ is the known heterogeneous variance term. The variance $s_i^2$ can be regarded as the inverse weight for each observation, and for the homogeneous setting $s_i^2 = 1$. The model formulation can be equivalently expressed in a matrix-vector form as

$$
\begin{aligned}
\boldsymbol{y}|\boldsymbol{\beta} &\sim N(\boldsymbol{\beta}, \sigma^2 S), \\
\beta_1 &\sim C(\cdot), \\
(D\boldsymbol{\beta})_j &\sim g(\cdot),
\end{aligned}
\tag{3}
$$

where $\boldsymbol{y} = (y_i : i = 1, 2, ..., n)$, $\boldsymbol{\beta} = (\beta_i : i = 1, 2, ..., n)$, $S = \mathrm{diag}(s_i^2)$, and $D$ is the first order difference matrix defined as equation 1.

## 2.2 Choice of prior

We choose a shrinkage prior $g(\cdot)$ such that the difference between two neighboring signals are mostly small (close to 0), which would lead to a spatially-structured signal. In our model formulation, the choices of shrinkage priors are flexible. In this paper, we focus on mixtures of normal distributions including the point-normal and adaptive shrinkage (ash) prior (Stephens, 2017). The prior is represented as

$$
g = \sum_{k=1}^{K} \pi_k N(0, \sigma_k^2 \sigma^2),
\tag{4}
$$

where $\sum_k \pi_k = 1$, and $\sigma^2$ is the random error variance in model equation 2. For point-normal prior (spike and slab prior with normal components), $K = 2$, and $\sigma_1^2$ is often fixed at 0 or a very small number, while $\{\pi_1, \sigma_2^2\}$ are the hyperparameters. For ash prior, $K$ is often large, and all $\sigma_k^2$ are known and fixed, and span a large grid (from a small value to large ones), while $\pi_k$'s are the hyperparameters. In this paper, we make a novel extension of the ash prior such that $\sigma_k^2$ are not fixed excepting the first one (the one for the spike component). Both priors have been applied to mean estimation and inference (Castillo & Roquain, 2020; Willwerscheid & Stephens, 2021), matrix factorization (Ning & Ning, 2024; Wang & Stephens, 2021), sparse regression (Kim et al., 2024; Ray & Szabó, 2022) and wavelet denoising (Chipman et al., 1997; Xing et al., 2021), and they have been shown to have better performance over other choices of priors.

## 2.3 The variational algorithm

For hyperparameters in the prior, we can either fix them before model fitting, or learn them from data. In this paper, we take the empirical Bayes approach for estimating the prior, and the posterior inference is conditional on the estimated prior distribution. While it's possible to use MCMC for sampling from the posterior, it's intractable for large-scale datasets (Quiroz et al., 2019). We instead propose to use variational inference for posterior computation. We start this section with a high-level review of empirical Bayes and variational inference.

### 2.3.1 Review of empirical Bayes and variational inference

An empirical Bayes (EB) approach estimates the prior by maximizing the marginal likelihood $p(\boldsymbol{y}; g) = \int p(\boldsymbol{y}|\boldsymbol{\beta})g(\boldsymbol{\beta})d\boldsymbol{\beta}$, then the posterior is computed conditional on $\hat{g}$ as $p(\boldsymbol{\beta}|\boldsymbol{y}, \hat{g})$. EB has been widely applied in large-scale multiple testing (Efron, 2004; Stephens, 2017; Liu et al., 2024; Xie, 2025), shrinkage estimation (Johnstone & Silverman, 2004; Koenker & Mizera, 2014; Dey et al., 2018), and other tasks such as sparse regression and model selection (Martin et al., 2017; Kim et al., 2024; Zou et al., 2024).

Variational Inference (VI, Blei et al. (2017)) turns the posterior inference problem into an optimization problem by approximating the true posterior with a more tractable distribution. The VI finds

$$
q^*(\boldsymbol{\beta}) = \arg\min_{q \in \mathcal{Q}} D_{KL}(q(\boldsymbol{\beta}) \| p(\boldsymbol{\beta}|\boldsymbol{y}; g)),
$$

where $\mathcal{Q}$ is a family of approximate densities, and $D_{KL}$ is the Kullback-Leibler (KL) divergence. In practice, we maximize the Evidence Lower Bound (ELBO), which is a lower bound on the $\log p(\boldsymbol{y}; g)$:

$$
\begin{aligned}
F(q; g, \boldsymbol{y}) &= \log p(\boldsymbol{y}; g) - D_{KL}(q(\boldsymbol{\beta}) \| p(\boldsymbol{\beta}|\boldsymbol{y})), \\
&= \mathbb{E}_{q(\boldsymbol{\beta})}(\log p(\boldsymbol{y}, \boldsymbol{\beta}; g) - \log q(\boldsymbol{\beta})).
\end{aligned}
$$

Variational empirical Bayes (VEB) combines variational inference and empirical Bayes in a single optimization problem, expressed as

$$
q^*(\boldsymbol{\beta}), \hat{g} = \underset{q \in \mathcal{Q}, g \in \mathcal{G}}{\arg\max} F(q, g; \boldsymbol{y}).
$$

### 2.3.2 Variational empirical Bayes for trend filtering

We develop the variational inference algorithm for model equation 2 in this section. For the prior in equation 4, we follow the standard approach for the Gaussian mixture model and introduce the latent variable $z$ such that

$$
\begin{aligned}
(\beta_{j+1} - \beta_j)|z_{jk} = 1 &\sim N(0, \sigma_k^2 \sigma^2), \\
p(z_{jk} = 1) &= \pi_k, \text{ for } j = 1, 2, ..., n - 1.
\end{aligned} \tag{5}
$$

For the variational posterior, we consider the following variational distribution class that factorizes over $\boldsymbol{\beta}$ and $\boldsymbol{z}$:

$$
q(\boldsymbol{\beta}, \boldsymbol{z}) = q_\beta(\boldsymbol{\beta}) q_z(\boldsymbol{z}) = N(\boldsymbol{\beta}; \bar{\boldsymbol{\beta}}, V) \prod_{j=1}^{n-1} \prod_{k=1}^{K} \alpha_{jk}^{z_{jk}}, \tag{6}
$$

where $\alpha_{jk} = q_{z_{jk}}(z_{jk} = 1)$ is the posterior probability that $(\beta_{j+1} - \beta_j)$ is drawn from the $k$th mixture component. When $V$ is a diagonal matrix, the posterior distribution is fully factorized. But we do not make such simplification and assume $V$ is a general covariance matrix. The evidence lower bound for model equation 2 is then

$$
F_{\text{EBTF}} = \mathbb{E}_q \log p(\boldsymbol{y}|\boldsymbol{\beta}; \sigma^2) + \mathbb{E}_q \log p(\boldsymbol{\beta}, \boldsymbol{z}; \boldsymbol{\pi}, (\sigma_k^2)) - \mathbb{E}_q \log q(\boldsymbol{\beta}, \boldsymbol{z}). \tag{7}
$$

For the optimization of the ELBO, we take the VEB approach introduced in the section 2.3.1 – since $\sigma^2$, $\boldsymbol{\pi}$, and prior variances $(\sigma_k^2)$ are all unknown, we treat them as parameters, which are optimized when maximizing the ELBO. The coordinate ascent algorithm for fitting EBTF model equation 2 has the following updates:

1. Given $q_\beta$, the update of the posterior probabilities $\alpha_{jk}$ is

$$
\alpha_{jk} = \frac{\pi_k N((D\bar{\boldsymbol{\beta}})_j^2 + (DVD^T)_{jj}; 0, \sigma^2 \sigma_k^2)}{\sum_{l=1}^{K} \pi_l N((D\bar{\boldsymbol{\beta}})_j^2 + (DVD^T)_{jj}; 0, \sigma^2 \sigma_l^2)}.
$$

2. Given $q_z$, the update for posterior variance $V$ and posterior mean $\bar{\boldsymbol{\beta}}$ are

$$
V = \sigma^2 (S^{-1} + D^T W D)^{-1}, \quad \bar{\boldsymbol{\beta}} = Vy/\sigma^2.
$$

3. Given $q_\beta, q_z$, the update for the prior variances $\sigma_k^2$, and prior probabilities $\boldsymbol{\pi}$ are

$$
\sigma_k^2 = \frac{\sum_j \alpha_{jk}((D\bar{\boldsymbol{\beta}})_j^2 + (DVD^T)_{jj})}{\sigma^2 \sum_j \alpha_{jk}}, \quad \pi_k \propto \sum_j \alpha_{jk}.
$$

4. Given the rest, let $\Omega = S^{-1} + D^T W D$, update $\sigma^2$ as

$$
\sigma^2 = (y^T S^{-1} y - 2y^T S^{-1} \bar{\boldsymbol{\beta}} + \bar{\boldsymbol{\beta}}^T \Omega \bar{\boldsymbol{\beta}} + \text{tr}(\Omega V))/(2n - 1).
$$

---

**Algorithm 1** VEB algorithm for fitting EBTF equation 2 (outline only)

---

    **Input:** Data $y_i$, variances $s_i^2$, for $i = 1, 2, ..., n$.

    **Init:** Posterior mean $\bar{\boldsymbol{\beta}}$, posterior precision matrix diagonal $\boldsymbol{d}$ and super-diagonal $\boldsymbol{e}$, residual variance $\sigma^2$, prior weights $\pi_k$ and variances $\sigma_k^2$ for $k = 1, 2, ..., K$.

    **repeat**

        1. Update posterior weights $\alpha_{ik}$ for $i = 2, 3, ..., n$ and $k = 1, 2, ..., K$ ;

        2. Update posterior precision matrix (its diagonal and super-diagonal elements only);

        3. Update $\bar{\boldsymbol{\beta}}$ by solving a (tridiagonal) banded linear system;

        4. Update prior weights, variances, and residual variance.

    **until** converged

---

The posterior precision matrix $V^{-1} = (S^{-1} + D^T W D)/\sigma^2$ is a tridiagonal matrix, because of the tridiagonal structure of the difference matrix $D$. This indicates that the sequences $(\beta_i)$ are conditionally independent a posteriori given two adjacent variables. Specifically, for the posterior distribution $q_{\beta_i}$, the adjacent two neighbors $q_{\beta_{i-1}}$ and $q_{\beta_{i+1}}$ are all the information needed to determine $q_{\beta_i}$.

Although the updates are formulated in matrix multiplication form, the computation cost can be significantly reduced by leveraging the special structure of the difference matrix. The matrix $D^T W D$ is tridiagonal and can be calculated fast by operations only on $w_i$, yielding its diagonal and super-diagonal elements. An optimized banded system solver (such as `scipy.linalg.solveh_banded`) can be used to find $\bar{\boldsymbol{\beta}}$. To find the diagonal of $DVD^T$, the diagonal and super-diagonal elements of $V$ are first obtained by inverting the tridiagonal precision matrix using the recursion algorithm from Usmani (1994). Then $DVD^T$ can be directly calculated using operations only on the diagonal and super-diagonal elements of $V$. The final algorithm for implementation is summarized in Algorithm 1. The algorithm iteratively solves for the maximum of each parameter while keeping all other parameters fixed, ensuring that every update increases the objective function.

*Remark* 2.1. In variational inference, the ELBO is in general not convex, and the initialization is important for non-convex optimization problems. However, there are fast initialization methods that can provide a good starting point for the variational algorithm. We address these initialization issues here. The residual variance $\sigma^2$ is initialized by applying median absolute deviation (MAD) to the finest level of wavelet coefficients as described in section 4.2 of Donoho & Johnstone (1994). For heterogeneous variances where $s_i^2$ are unknown, we may use the running MAD estimator proposed in Gao (1997) or the wavelet-based variance estimation in Xing et al. (2021). The precision matrix is initialized to the identity matrix. The posterior mean $\bar{\boldsymbol{\beta}}$ is initialized to the wavelet denoised mean by applying soft thresholding at $\sigma\sqrt{2\log n}$ to the Haar wavelet coefficients. The wavelet method is chosen because it is fast and the Haar wavelet threshold provides piecewise constant signal estimation. An alternative initialization for the posterior mean $\bar{\boldsymbol{\beta}}$ is cross-validated trend filtering, which is generally scalable and provides a reasonably good starting value. We compare the two strategies in the simulation study and find that they yield similar results, as indicated by the comparable RMSEs shown in Figure 11 in Appendix A. In contrast, if $\bar{\boldsymbol{\beta}}$ is initialized at the data points $\boldsymbol{y}$, the method typically fails to converge, causing the learned curve to remain stuck at $\boldsymbol{y}$.

# 3 Alternative formulations of the EB trend filtering

In this section, we present two alternative formulations of the EBTF model, and show the equivalence among all formulations in terms of the objective function ELBO. We further discuss the pros and cons for each formulation.

### 3.1 Multivariate normal variance prior formulation

The primary model equation 2 can be formulated in an equivalent way as

$$\begin{aligned}
\boldsymbol{y}|\boldsymbol{\beta} &\sim N(\boldsymbol{\beta}, \sigma^2 S), \\
D\boldsymbol{\beta}|W &\sim N(0, \sigma^2 W), \\
W_{jj} &\sim \tilde{g}(\cdot), \text{ for } j = 1, 2, ..., n-1,
\end{aligned} \tag{8}$$

where $W$ is a diagonal covariance matrix $W = \text{diag}(w_j : j = 1, 2, ..., n-1)$, and $\tilde{g}$ is a prior on the variances. Specifically, the $\tilde{g}$ corresponding to the prior equation 4 is $w_j \sim \text{Discrete}(\sigma_1^2, ..., \sigma_k^2; \boldsymbol{\pi})$, where the discrete distribution is defined as $p(w_j = \sigma_k^2) = \pi_k$, for $k = 1, 2, ..., K$, with $\sum_k \pi_k = 1$. This formulation is named the multivariate normal variance (MNV) prior approach, as it introduces a multivariate normal prior on $D\boldsymbol{\beta}$, followed by another prior on the variances. We use the VEB framework for prior estimation and posterior computation by maximizing the ELBO

$$F_{\text{MNV}} = \mathbb{E}_q \log p(\boldsymbol{y}, \boldsymbol{\beta}, W; g) - \mathbb{E}_q \log q(\boldsymbol{\beta}, W).$$

The VEB updates for maximizing the ELBO $F_{\text{MNV}}$ are the same as those in Section 2.3.2 for fitting the EBTF model equation 2, if the variational posterior distribution for model equation 8 is chosen as

$$q(\boldsymbol{\beta}, W) = q_{\boldsymbol{\beta}}(\cdot) q_W(\cdot) = N(\boldsymbol{\beta}; \bar{\boldsymbol{\beta}}, V) \prod_j q_{w_j}(\cdot). \tag{9}$$

Although the VEB updates remain the same, the VEB algorithm 2 (in Appendix C) of the MNV approach has a nice property: it alters between a simple update on $q_{\boldsymbol{\beta}}$, and a general empirical Bayes Gaussian variance (EBGV, see Appendix B) problem for $(\tilde{g}, q_W)$. Hence, the MNV model formulation and inference are modular. To accommodate different prior distributions on $W$, it is sufficient to develop the corresponding EBGV problem for these priors, instead of re-deriving the full variational updates. The EBGV solver can then be plugged into the general variational inference iterations.

### 3.2 Multiple linear regression formulation

The trend filtering problem can be formulated as a penalized multiple linear regression (MLR) problem, as shown in Lemma 2 of Tibshirani (2014) . Specifically, let $H \in \mathbb{R}^{n \times n}$ be the "inverse" of the first-order difference matrix $D$, such that $D\boldsymbol{\beta} = \boldsymbol{b}, H\tilde{\boldsymbol{b}} = \boldsymbol{\beta}$, where $\tilde{\boldsymbol{b}} = (\beta_1, \boldsymbol{b}^T)^T$.

The first element of $\tilde{\boldsymbol{b}}$ is $\beta_1$, which can be regarded as the baseline value, and all the subsequent signals are additions or subtractions to it. The vector $\boldsymbol{b}$ captures the piecewise difference among the remaining signals. Thus, model equation 2 can be reformulated as a Bayesian sparse multiple linear regression problem:

$$\begin{aligned}
\boldsymbol{y}|\tilde{\boldsymbol{b}} &\sim N(H\tilde{\boldsymbol{b}}, \sigma^2 S), \\
\beta_1 &\sim C(\cdot), \\
b_j &\sim g(\cdot) \text{ for } j = 1, 2, ..., n-1,
\end{aligned} \tag{10}$$

where the prior $g(\cdot)$ is sparsity-inducing and is the same as equation 4.

If the variational posterior distribution for model equation 10 is chosen as

$$q(\tilde{\boldsymbol{b}}, \boldsymbol{z}) = q_{\tilde{\boldsymbol{b}}}(\tilde{\boldsymbol{b}}) q_z(\boldsymbol{z}) = N(\tilde{\boldsymbol{b}}; \bar{\bar{\boldsymbol{b}}}, V_{\tilde{b}}) \prod_{j=1}^{n-1} \prod_{k=1}^{K} \alpha_{jk}^{z_{jk}}, \tag{11}$$

then the ELBO for the the multiple linear regression formulation is

$$F_{\text{MLR}} = \mathbb{E}_q \log p(\boldsymbol{y}|\tilde{\boldsymbol{b}}) + \mathbb{E}_q \log p(\tilde{\boldsymbol{b}}, \boldsymbol{z}) - \mathbb{E}_q \log q_{\tilde{\boldsymbol{b}}}(\tilde{\boldsymbol{b}}) - \mathbb{E}_q \log q_z(\boldsymbol{z}),$$

and $F_{\text{MLR}}$ is equivalent to the ELBO equation 7 for the primary model formulation equation 2. Hence the VEB updates for both models are the same.

Given the equivalence of the three model formulations equation 2, equation 8 and equation 10, we comment on their advantages and disadvantages. The multiple linear regression formulation is very general, and there is a large number of methods for Bayesian sparse linear regression. Hence those methods can be readily applied to Bayesian trend filtering with minimal modifications (though the availability of EB sparse regression methods is limited). The multivariate normal variance prior formulation is modular and can easily incorporate different types of priors.

However, from a modeling perspective, the MNV and MLR approaches are specific to the pre-defined trend filtering problem and are not easily generalized to more sophisticated models. On the other hand, the primary dynamic linear model formulation of the trend filtering is more flexible in terms of adding additional model components and adding custom features. In Section 5, we illustrate this perspective by constructing and solving a sparse and spatially-structured sequencing model. See the related discussion in Section 7.2 of Tibshirani (2014).

## 4 Numerical Examples

In this section, we evaluate the performance of our proposed method, EBTF, against other widely used non-parametric regression methods. All experiments are conducted on a Linux system with an i9-10900F processor and 32GB memory. The compared methods are:

1. genlasso-tf: cross-validated zeroth-order trend filtering using the `R` function `cv.trendfilter` from the `genlasso` package (Arnold & Tibshirani, 2014).

2. wave-hard and wave-Bayes: Haar wavelet denoising (hard thresholding, (Donoho & Johnstone, 1994)) and Bayesian adaptive shrinkage (Chang et al., 2000)) using the `Python` function `denoise_wavelet` from the `skimage` package (Van der Walt et al., 2014).

3. susie-tf: an empirical Bayes variable selection method extended for trend filtering, using the `R` function `susie_trendfilter` from the `susieR` package (Wang et al., 2020). We consider three settings where $L = 10, 20, 30$ in the simulation.

4. BTF: Bayesian trend filtering with dynamic horseshoe (DHS) prior and normal-inverse-gamma prior (NIG) from the `R` package `dsp` (Kowal et al., 2019). We use the default setting with 1000 MCMC burn-in iterations and 4000 final MCMC iterations for posterior calculations. We have found that setting $D = 0$ (zeroth-order) generally leads to poor fitting and convergence issues, so we set $D = 1$ in the `btf` function.

5. GP: Gaussian process regression with the Matern 3/2 kernel using the `GaussianProcessRegressor` from `scikit-learn`. When fitting the Gaussian process, we first standardize the data before model fitting and then convert the fitted mean back to the original scale. As noted by one of the reviewers, this approach improves performance and speeds up GP runtime.

### 4.1 Simulation

We consider six different signal functions: blocks, steps, bumps, Gaussian density (Gauss), linear, and Heavisine. These functions are illustrated in Figure 8. We set the number of samples to be $n = 1024$, and the residual variance to $\sigma^2 = 1$. The signal-to-noise ratio (SNR) is defined as $\text{SNR} = \text{Var}(\boldsymbol{\beta})/\sigma^2$ and is set to 3. Each experiment is repeated 20 times, and we report the averaged root mean squared error (RMSE), mean absolute error (MAE) and the coefficient of determination ($R^2$) between the estimated and the true signal along with their standard errors. The metrics are defined as

$$\text{RMSE} = \sqrt{\frac{1}{n}||\boldsymbol{\beta} - \hat{\boldsymbol{\beta}}||_2^2}, \ \ \text{MAE} = \frac{1}{n}|\boldsymbol{\beta} - \hat{\boldsymbol{\beta}}|_1, \ \ R^2 = 1 - \frac{\sum_{i=1}^n (y_i - \hat{y}_i)^2}{\sum_{i=1}^n (y_i - \bar{y})^2},$$

where for Bayesian methods, $\hat{\boldsymbol{\beta}}$ is the posterior mean.

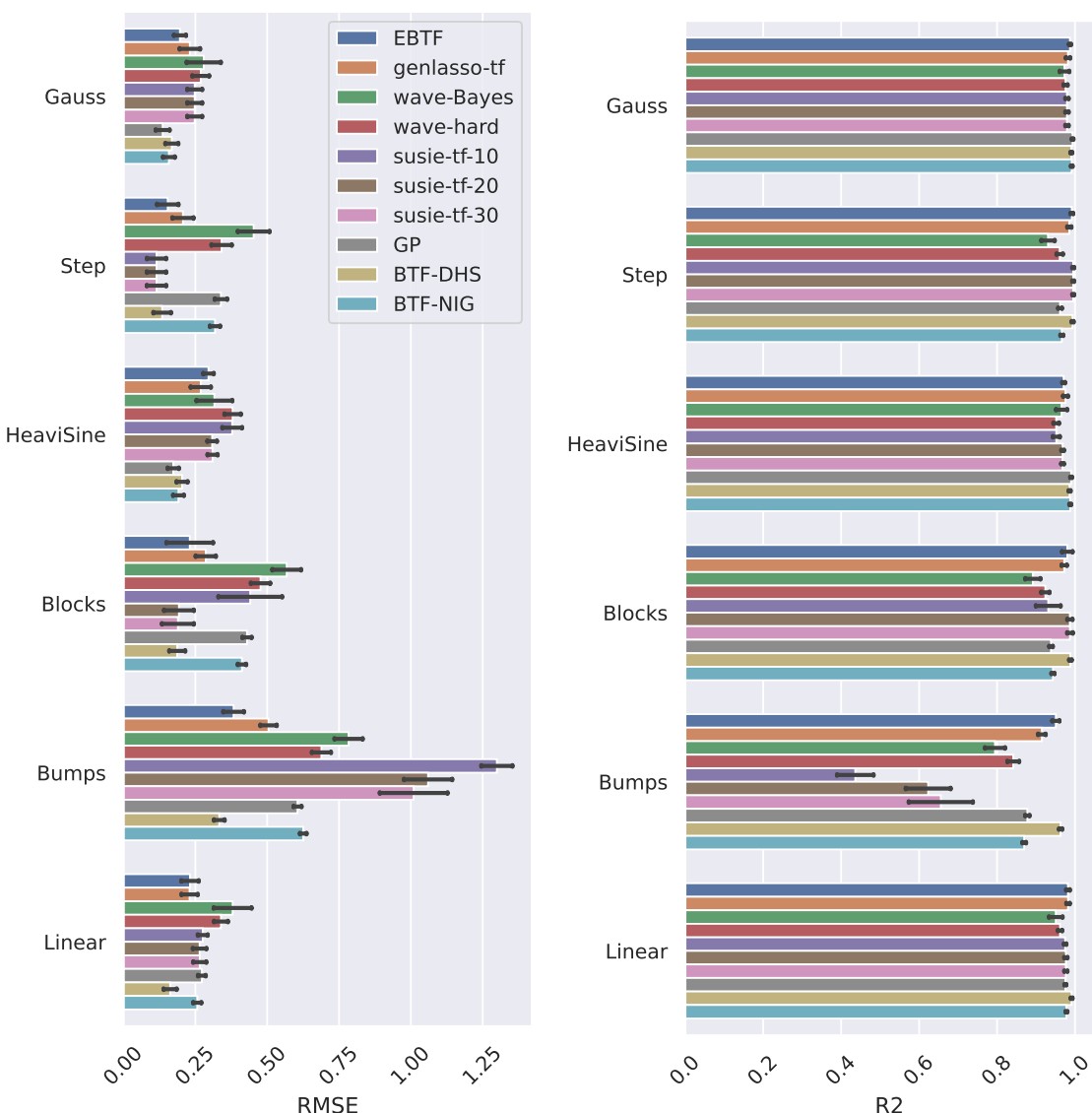

Figure 1: Simulation results: RMSE and $R^2$ scores of the competing methods across six signal functions. Each bar represents the metric value, while the horizontal lines on the bars indicate the standard error over 20 replications. The left panel presents the Root Mean Squared Error (RMSE), where lower values indicate better performance, while the right panel shows the $R^2$ metric, where higher values indicate better fit. Different colors correspond to different signal functions, as shown in the legend. The results for the Mean Absolute Error (MAE) follow a similar trend to RMSE and are provided in Figure 9 in Appendix A.

Figure 1 presents the RMSE and $R^2$ scores for all methods across different signal functions. The proposed EBTF method generally achieves the near-lowest RMSE and the near-highest $R^2$ scores. Other strong performers include BTF-DHS, genlasso-tf, wavelet denoising with hard thresholding, and Gaussian process, which all demonstrate robust performance across various signal functions. Notably, GP performs best on the Gauss and Heavisine signals, benefiting from their smooth nature. The BTF with DHS prior achieves the lowest RMSE in a couple of signal functions – blocks, bumps and linear. The DHS prior also consistently outperforms the NIG prior in BTF. EBTF and BTF-DHS have consistently high $R^2$ scores in all the signal functions, particularly for the bumps function.

The susie-tf-30 method provides the best estimations for blocks and step signals, while EBTF, BTF-DHS also perform well on these piecewise constant functions. Increasing the $L$ parameter in susie generally improves performance, particularly from $L = 10$ to $L = 20$, although further increasing it to $L = 30$ yields only marginal gains. However, susie-tf struggles with the bumps function, showing severe underfitting as the fitted curve is often missing most of the bumps.

Figure 2 provides a high-level summary of the simulations, illustrating the trade-off between estimation accuracy and runtime. Wavelet-based methods are significantly faster than the rest of methods, as they leverage the efficient pyramid algorithm for wavelet decomposition and reconstruction. EBTF offers consistently strong performance while maintaining a fast runtime; its overall $R^2$ is only slightly smaller than BTF-DHS yet it runs faster than almost all methods except wavelet-based approaches. While susie-tf-20 achieves lower RMSE than susie-tf-10 and wavelet methods, this comes at the cost of substantially higher runtime. Increasing susie's $L$ from 20 to 30 yield slightly performance gains but also results in an increase in computational cost. BTF-DHS achieves the lowest RMSE and highest $R^2$ scores overall but at the expense of significantly longer runtime. In contrast, BTF-NIG has a runtime comparable to susie-tf-20 but with lower performance.

## 4.2 Real data

In this section, we show the applications of EBTF to several real datasets. The first dataset, motorcycle acceleration (Silverman, 1985), is a classical example used for illustrating the nonparametric regression methods. It provides measurements of head acceleration in a simulated motorcycle accident, used to test crash helmets. For comparison, we added fitted curves from the other methods (for susie-tf we used $L = 20$). The black curve in Figure 3 is the EBTF fit, and clearly it captures the trend of the acceleration over time. On the other hand, both the genlasso-tf and susie-tf exhibit some degree of under-fitting. The genlasso-tf seems to overshrink the signal around time 60 and time 100. It also underestimates the signal around time 0 to 20, as the estimated signal is clearly below all the observations in that time period. The susie-tf seems to underfit the signal in the time period from 70 to 90, as it only produces one big jump there. The GP provides a smooth overall fit and captures the acceleration trend, while the wavelet-based method exhibits abrupt jumps and underfits the region around time period 60.

We next evaluate the performance of the methods on datasets with longer sequences and more abrupt changes in trends. The three datasets used in our experiments, sourced from Wu et al. (2021), are described as follows:

- **ETTh1 dataset:** This dataset consists of time-series data collected from electricity transformers, including oil temperature, recorded at an hourly frequency between July 2016 and July 2018. The total sequence length is 17420.

- **Illness dataset:** This dataset contains 966 weekly records of influenza-like illness (ILI) cases in the United States between 2002 and 2021. It captures the ratio of ILI patients to the total number of patients seen.

- **Weather dataset:** This dataset includes $CO_2$ concentration measurements recorded every 10 minutes throughout 2020. The dataset has more than 50000 records.

To visualize the fitted results for the ETTh1 dataset, we select three 500-length segments, as shown in Figure 4. From Figure 4, we observe that genlasso-tf and susie-tf ($L = 500$) exhibit noticeable underfitting, while the EBTF, GP, and wavelet-based methods demonstrate better trend-capturing ability, with EBTF and GP

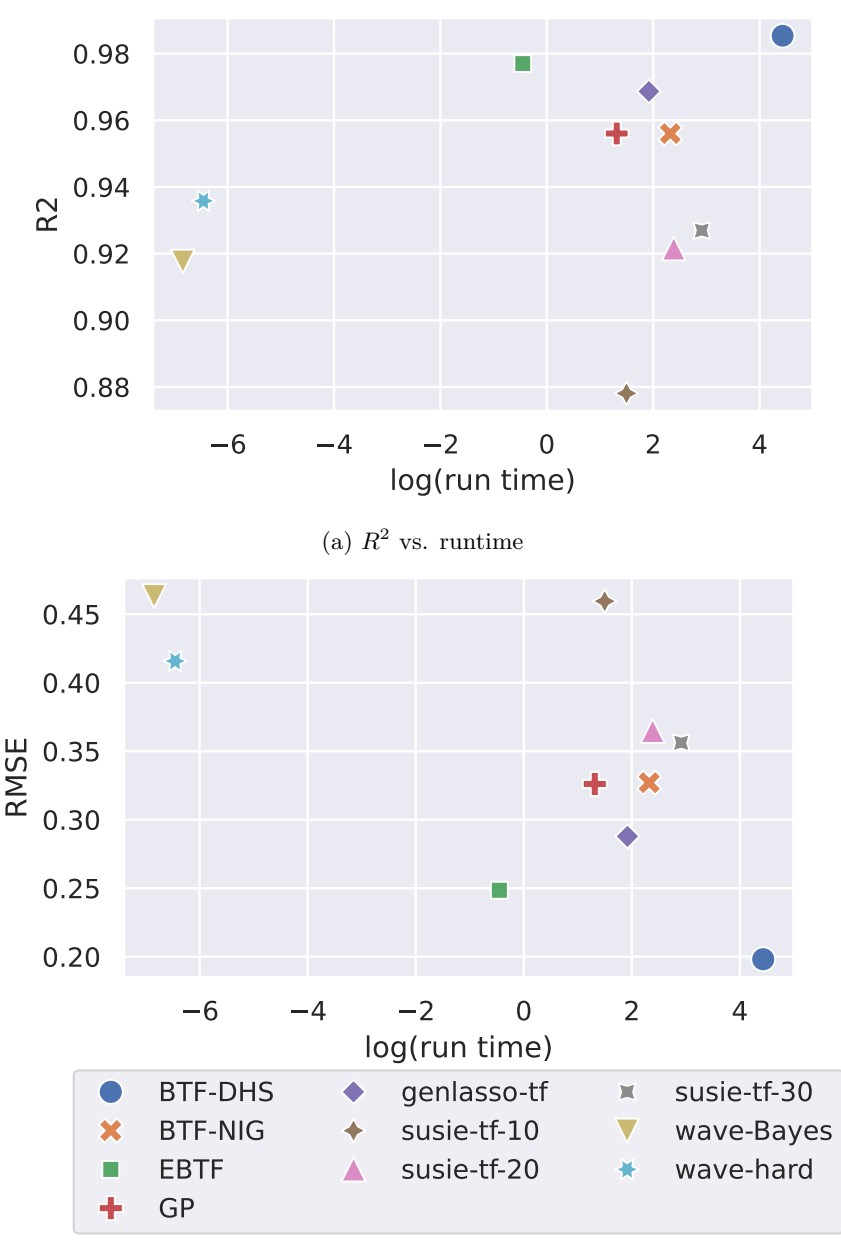

(a) $R^2$ vs. runtime

(b) RMSE vs. runtime

Figure 2: Simulation results: plots of log-transformed runtime vs performance metrics. The top panel (a) shows the relationship between log-transformed runtime and $R^2$ , while the bottom panel (b) displays log-transformed runtime versus RMSE. Each point represents a different method, and the legend indicates the corresponding method labels. The runtime is measured in seconds and log-transformed for better visualization. Both metrics and runtime values are averaged across all signal functions and repetitions for each method. The results for the MAE follow a similar trend to RMSE and are provided in Figure 10 in Appendix A

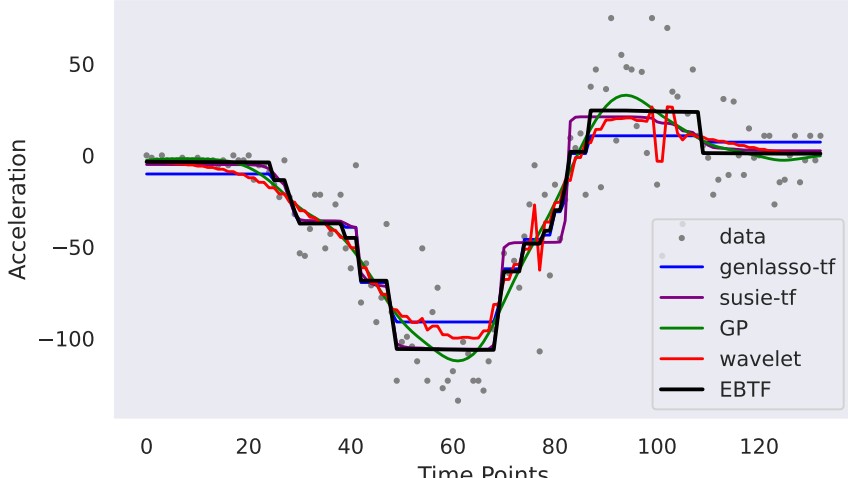

Figure 3: Motorcycle acceleration data from Silverman (1985).

capturing the most intricate trend details. EBTF and wavelet also appear to capture the trend of outliers, as shown in the top plot of Figure 4. Future work could explore improving robustness.

The fitted curves for the illness and weather datasets are presented in Figure 5. For the illness dataset, all methods capture finer seasonal variations, while susie-tf shows some underfitting at peaks, and wavelet exhibits abrupt jumps. The remaining three methods: EBTF, genlasso-tf, and GP, show very similar signal reconstructions. Similarly, susie-tf also underfits regions (such as around 700 and 900 minutes) with abrupt changes in the weather data. Increasing $L$ could potentially improve performance but at a significant computational cost. The other methods show very similar trends. Overall, the results highlight EBTF as an effective method for capturing both smooth and abrupt trend changes, while GP and wavelet-based methods provide reasonable approximations.

In addition, we evaluate the methods on holdout test data across the ETTh1, illness, and weather datasets. Specifically, for each dataset, we perform $K$-fold holdout with a structured fold assignment (Arnold & Tibshirani, 2014). In this setup, every $K$th data point is assigned to the same fold based on its sequential order, ensuring consistency across runs. The first and last points are excluded from all folds and always contribute to building the predictive model. The holdout set error is computed by averaging the predicted values from the two neighboring points of each test sample, which belong to different folds. This approach maintains the temporal structure of the data while providing stable and reproducible results.

We compare the RMSE ratio relative to a simple mean predictor (i.e., predicting using the mean of the observed data points) and present the results in Figure 6. Across all datasets and data sizes, EBTF, genlasso-tf, and wavelet-based methods achieve similar predictive performance, with relatively low RMSE ratios, and EBTF has the smallest RMSE when $n = 4096$. All methods perform well on the illness data, while on the weather data, GP and susie-tf have similar RMSE ratios that are slightly worse than the other methods when $n = 1024$.

Figure 6 also presents the log-scaled runtime comparison across methods. The wavelet-based method is the fastest, while EBTF demonstrates substantial runtime efficiency improvements over genlasso-tf, susie-tf, and GP, making it a competitive choice for large-scale time series analysis. Additionally, EBTF exhibits good scalability, as its runtime increases only marginally when increasing from $n = 1024$ to $n = 4096$, whereas GP shows a larger increase in computational cost. Despite susie-tf's underfitting issues, it remains the most computationally expensive method across all datasets. For longer sequences, susie-tf requires a larger $L$ to capture all trends effectively, but this comes at the expense of substantially increased runtime.

Overall, the results suggest that EBTF provides a strong balance between predictive accuracy and computational efficiency, outperforming GP and susie-tf in both accuracy and runtime while maintaining comparable

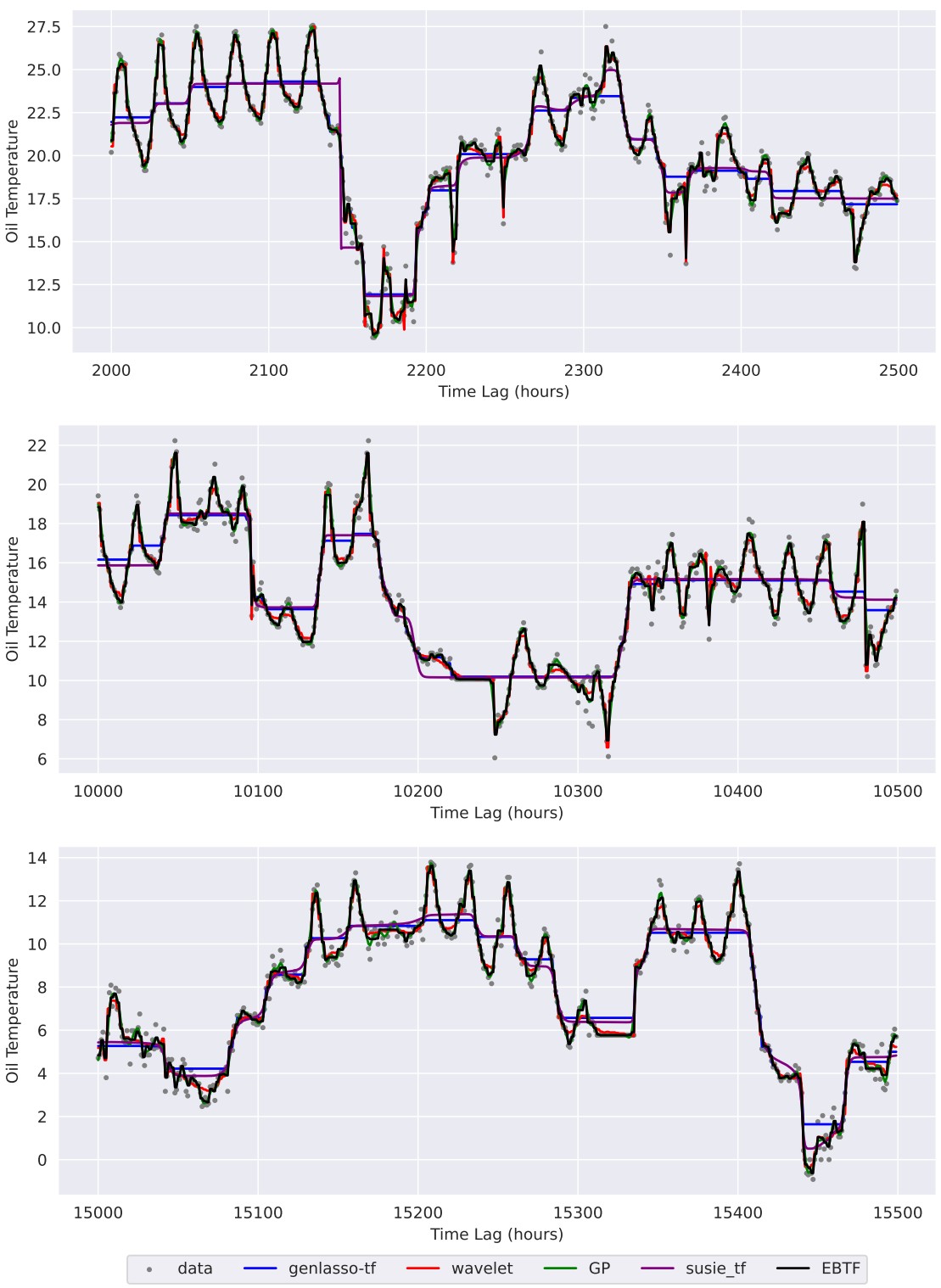

Figure 4: Fitted results for three selected 500-length segments from the ETTh1 dataset. The genlasso-tf and susie-tf ($L = 500$) methods underfit the trend, failing to capture the oscillatory wave patterns accurately. In contrast, EBTF captures most of the trend details, while GP and wavelet-based methods provide reasonable approximations.

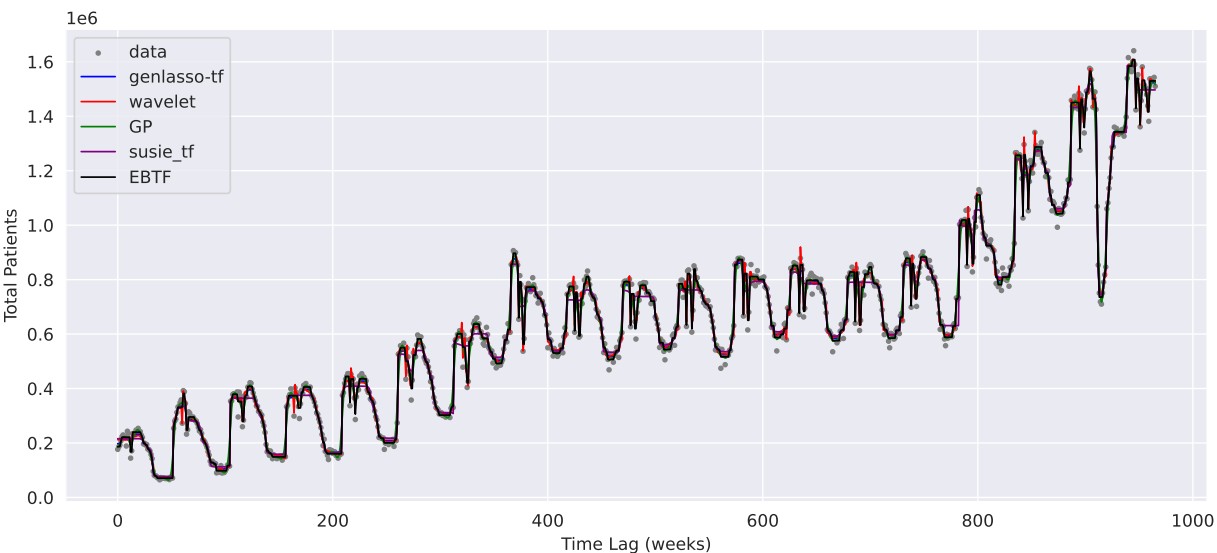

(a) The Illness dataset represents the number of total patients with ILI recorded weekly. All methods are able to capture finer seasonal variations, while susie-tf shows under-fitting at peaks and wavelet shows some abrupt jumps. The rest three methods have similar reconstruction of the signals.

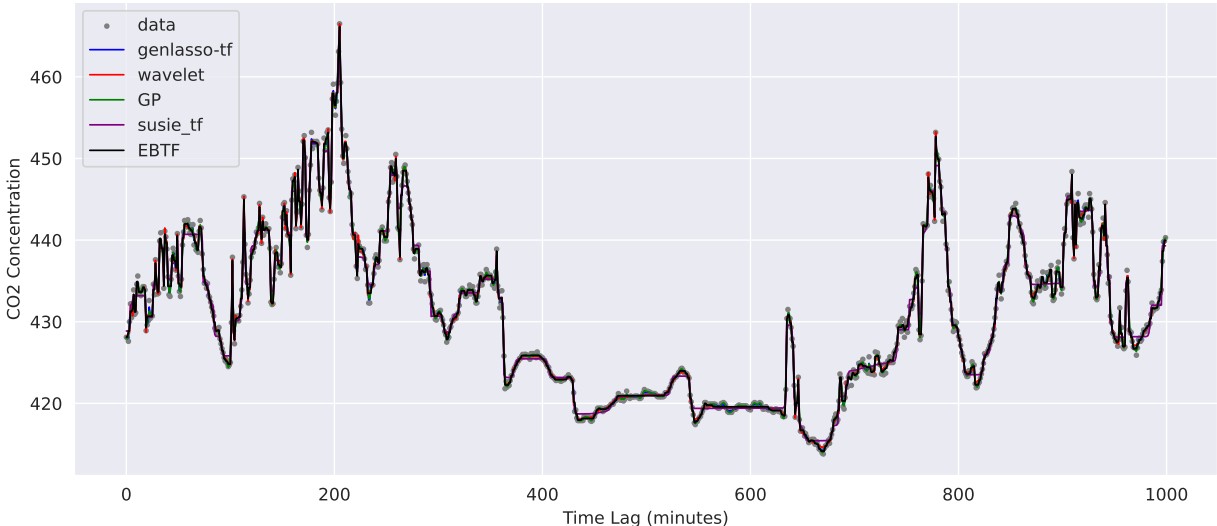

(b) The Weather dataset records $CO_2$ concentration every 10 minutes. We select the first 1000 data points to visualize. All methods, except susie-tf, effectively reconstruct both smooth and abrupt variations, while GP appears to better capture small fluctuations, particularly around the 600-minute mark.

Figure 5: Comparison of fitted trends for the Illness and Weather datasets.

performance to wavelet and genlasso-tf. Future work could explore optimizing EBTF further for even greater robustness and scalability.

Finally we applied the EBTF method to eight more real datasets, and the figures for the fitted signals are shown in Appendix A. The original data were processed by Van den Burg & Williams (2020), which evaluates several change-point detection methods. For comparison, we included genlasso-tf fitted signals (blue line) in all plots. Overall, EBTF (black line) provides more visually appealing signal fitting, especially in its ability to capture the data trend without overfitting.

## 5 Sparse Empirical Bayes Trend Filtering

The primary EBTF model (2) can be viewed as a general generative prior for a spatially-structured sequence and can be applied in various dynamic model settings. For example, inducing sparsity on the signal (Koop & Korobilis, 2023; Ramírez-Hassan, 2020; Rockova & McAlinn, 2021). A slight modification of the prior leads to a sparse empirical Bayes trend filtering (sparse EBTF) as follows:

$$
\begin{aligned}
\boldsymbol{y}|\boldsymbol{\beta} &\sim N(\boldsymbol{\beta}, \sigma^2 S), \\
\beta_1 &\sim \pi_0 N(0, \sigma^2 \sigma_0^2) + (1 - \pi_0) C(\cdot), \\
\beta_{j+1}|\beta_j &\sim \pi_0 N(0, \sigma^2 \sigma_0^2) + \sum_{k=1} \pi_k N(\beta_j, \sigma^2 \sigma_k^2), \text{ for } j = 1, 2, ..., n-1,
\end{aligned}
\tag{12}
$$

where $\sigma_0^2$ is a pre-chosen small variance value such that $N(0, \sigma^2 \sigma_0^2)$ is spiky, with $\sum_{k=0}^{K} \pi_k = 1$. We have added an extra mixture component that induces sparsity on the sequence $\beta_i$ directly. In particular, each element of the sequence is now a mixture of two components: one that promotes sparsity in $\beta_i$, and a smoothness-inducing component. The first component is a spiky normal distribution at 0 that shrinks $\beta_i$ towards 0, while the second one is the same as in the original trend filtering model formulation. For the posterior, we again consider the following variational distribution class that factorizes between $\boldsymbol{\beta}$ and $\boldsymbol{z}$, as

$$
q(\boldsymbol{\beta}, \boldsymbol{z}) = q_\beta(\boldsymbol{\beta}) q_z(\boldsymbol{z}) = N(\boldsymbol{\beta}; \bar{\boldsymbol{\beta}}, V) \prod_{i=1}^{n} \prod_{k=0}^{K} \alpha_{ik}^{z_{ik}},
\tag{13}
$$

where $\alpha_{ik} = q_{z_{ik}}(z_{ik} = 1)$ is the posterior probability indicating the mixture distribution. We define $\alpha_{1k} := 0$ for $k = 2, 3, ..., K$ since the prior of $\beta_1$ has only two mixture components. The detailed development of the VEB update for sparse EBTF is given in Appendix D.4.

To illustrate the effect of sparse EBTF on estimating sparse and spatially-structured signals, we present an example where sparse EBTF shrinks the sequence towards 0 when the underlying signals are truly sparse. We generated $n = 4096$ samples from the bump function, in which the signals are mostly at 0 and occasionally jump to large values. We fitted a sparse EBTF model to the data and compared the fit with the regular EBTF (without sparsity induction on the sequence). As shown in Figure 7, sparse EBTF is clearly able to shrink the estimated signals towards 0 while estimating the spatially-structured curve. However, without the signal-sparsity constraint, the estimated signals in sparse areas could be clearly non-zero, especially in areas between two spikes.

## 6 Extensions and Discussions

In this paper, we propose a fast and scalable empirical Bayes trend filtering method for nonparametric regression. The method leverages empirical Bayes estimation and variational inference, allowing it to learn the unknown smoothness level from the data. We demonstrated the superior performance of the EBTF method through simulations and real data examples. Our proposed variational posterior family is multivariate for the signal. This approach offers the benefit of fast computation while also maintaining the posterior dependency among all signals. An alternative posterior family is to factorize over observations but not over $\boldsymbol{\beta}, \boldsymbol{z}$, as $q(\boldsymbol{\beta}, \boldsymbol{z}) = \prod_i q(\beta_i|\boldsymbol{z}_i) q(\boldsymbol{z}_i)$. However this posterior assumes independence a posteriori and presents more computational challenge as we need to track the posterior of $\beta_i$ over $K$ components.

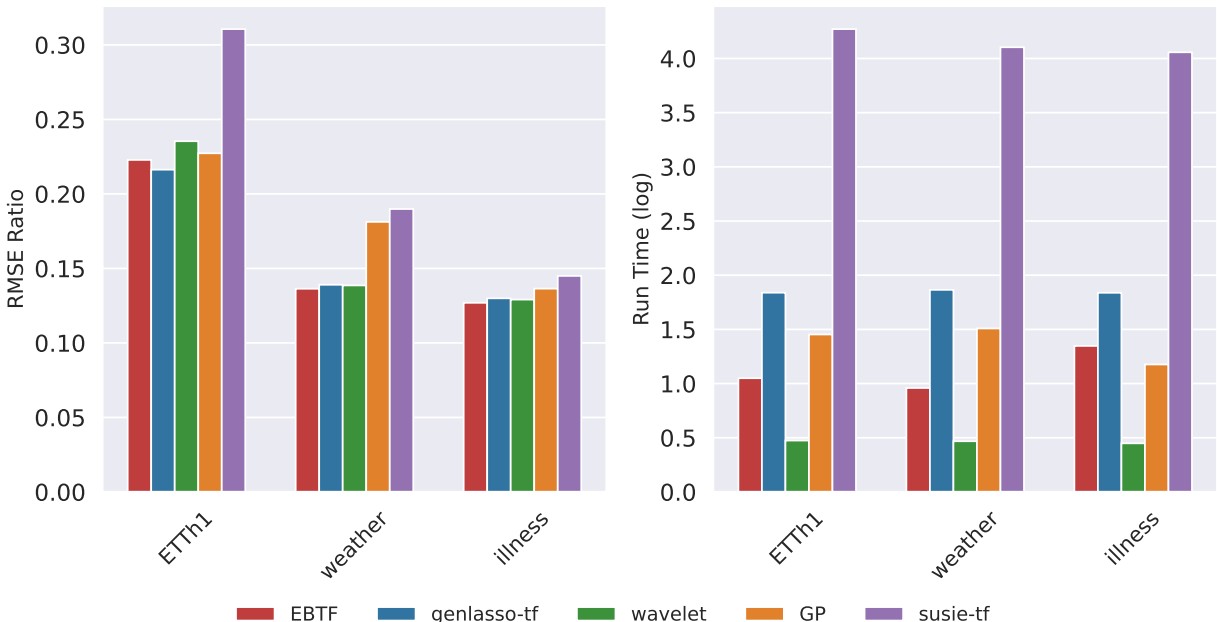

(a) Holdout evaluation results with $n = 1024$. The illness dataset has 966 records.

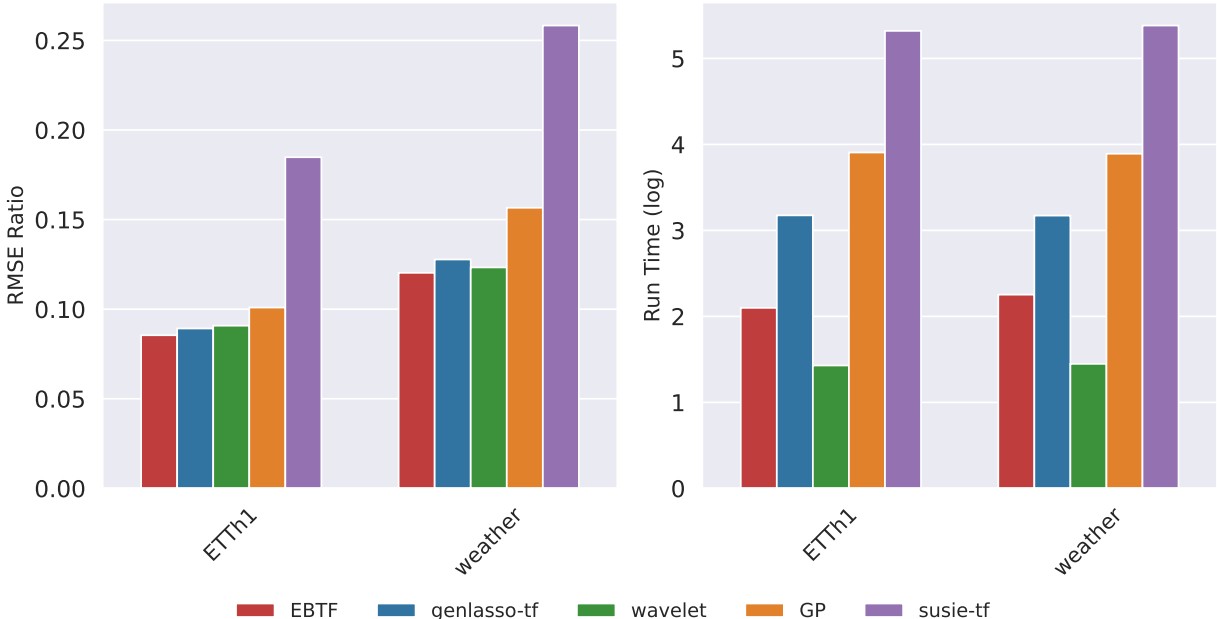

(b) Holdout evaluation results with $n = 4096$. The illness dataset has 966 records so it is not shown in this plot.

Figure 6: Comparison of RMSE ratio and runtime across ETTh1, weather, and illness datasets in the holdout test. For each dataset, we select the first $n = 1024$ data points (top figures) and $n = 4096$ data points (bottom figures) for evaluation, except for the illness dataset, which contains 966 records, and we use all available data. The left panel shows the RMSE ratio relative to a mean predictor, where lower values indicate better predictive performance. The right panel presents the log-scaled runtime comparison. EBTF achieves a balance between predictive accuracy and computational efficiency, demonstrating improvements over genlasso-tf, GP, and susie-tf.

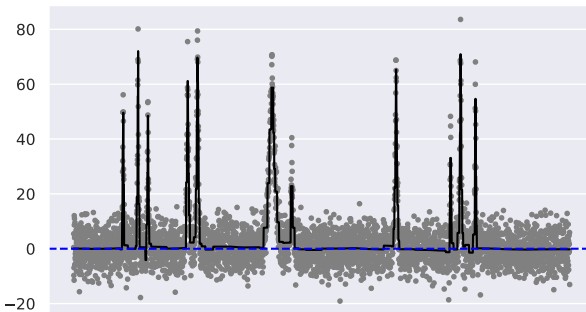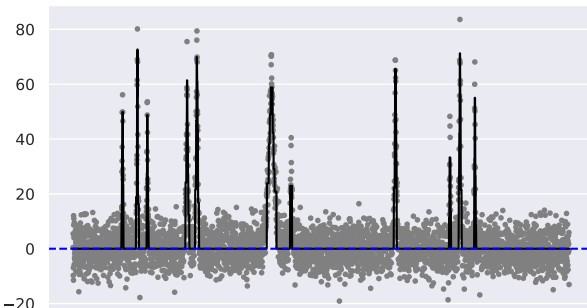

Figure 7: Illustration of sparse EBTF. The true signal is a bump function, as shown in Figure 8. This signal is mostly sparse at 0, while occasionally jumps to a large value. The left plot shows the regular EBTF fit, without inducing sparsity on the signals. The right plot shows the sparse EBTF fitted signal. The blue dashed line indicates $y = 0$.

In our approach, we have utilized a flexible non-parametric shrinkage prior on the differences. However, there are other Bayesian shrinkage priors that have been proposed recently, such as the widely used global-local shrinkage priors. Our flexible framework can easily incorporate different shrinkage priors, allowing for the selection of the appropriate prior based on the ELBO. The combination of shrinkage priors and variational inference presents an interesting direction for studying uncertainty quantification, particularly in the empirical Bayes setting (Xie & Stephens, 2022; Ignatiadis & Wager, 2022).

It is straightforward to extend our method to higher-order trend filtering by replacing the difference matrix $D$ with higher-order matrices, and all the results still hold. One difference is that when developing software implementations, we need to develop solvers and matrix manipulations for general banded matrices. For example, for $k = 1$, the posterior precision matrix is pentadiagonal.

As a final note, in real applications, the data may not always be real-valued: for non-Gaussian data, count and binary data are the two most commonly encountered types. For example in image denoising (Luisier et al., 2010), the pixel values are typically integers and we may assume they follow Poisson distribution. Variational inference methods have been developed for non-Gaussian likelihood by leveraging a Gaussian-based model, and our method can be easily adapted to handle these non-Gaussian data types. For example, see Seeger & Bouchard (2012); Xie (2023) for Poisson data, and Durante & Rigon (2019); Jaakkola & Jordan (1997) for binary data.

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

## A    Additional plots

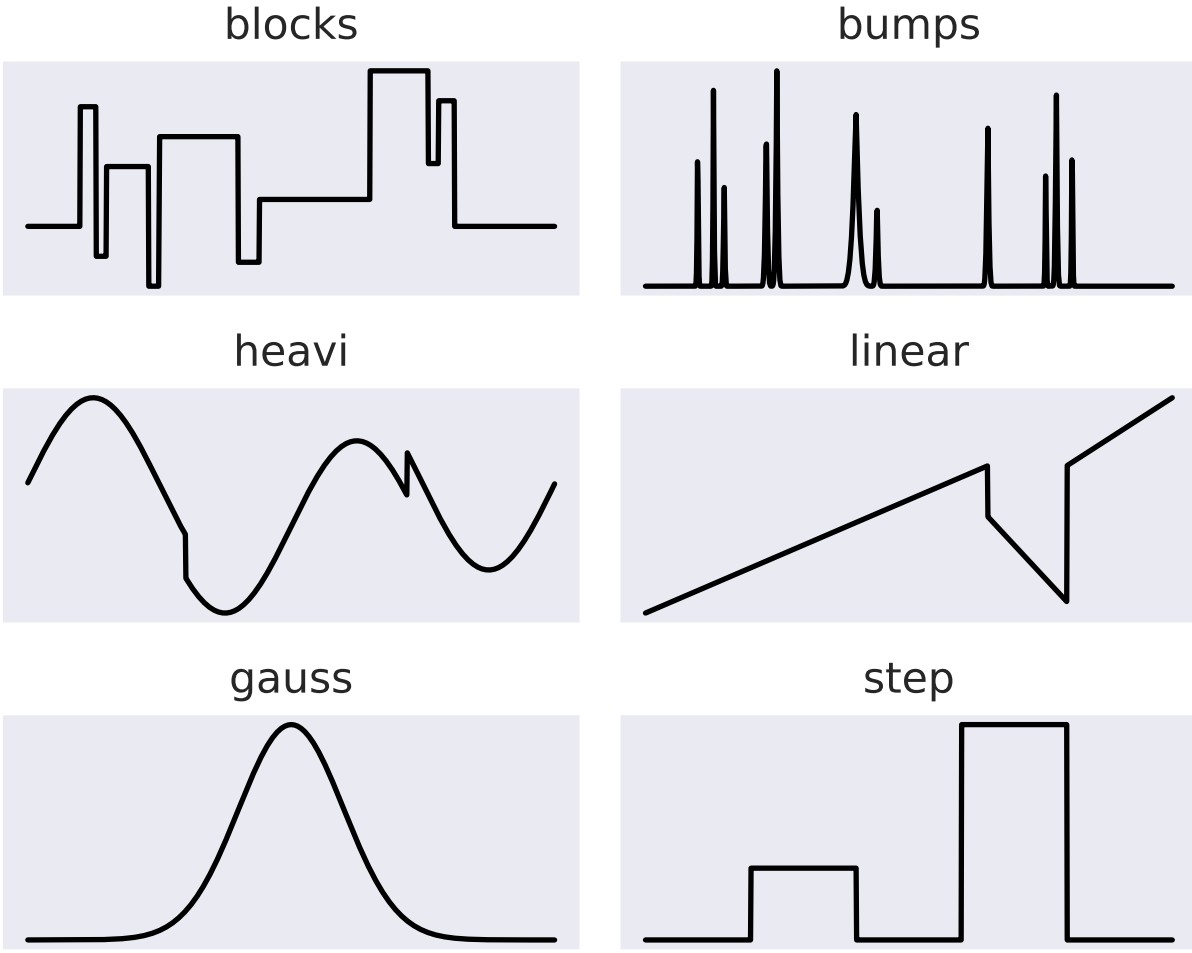

Figure 8: Six signal functions in the simulation study. The blocks, bumps and Heavisine functions are originally proposed by Donoho & Johnstone (1994) for evaluating wavelet denoising method. The blocks and step functions are piecewise constant; bumps have most signals at 0 but jump at certain locations; linear is a piecewise linear function; Heavisine is a piecewise twice differentiable function; Gauss is the Gaussian density function.

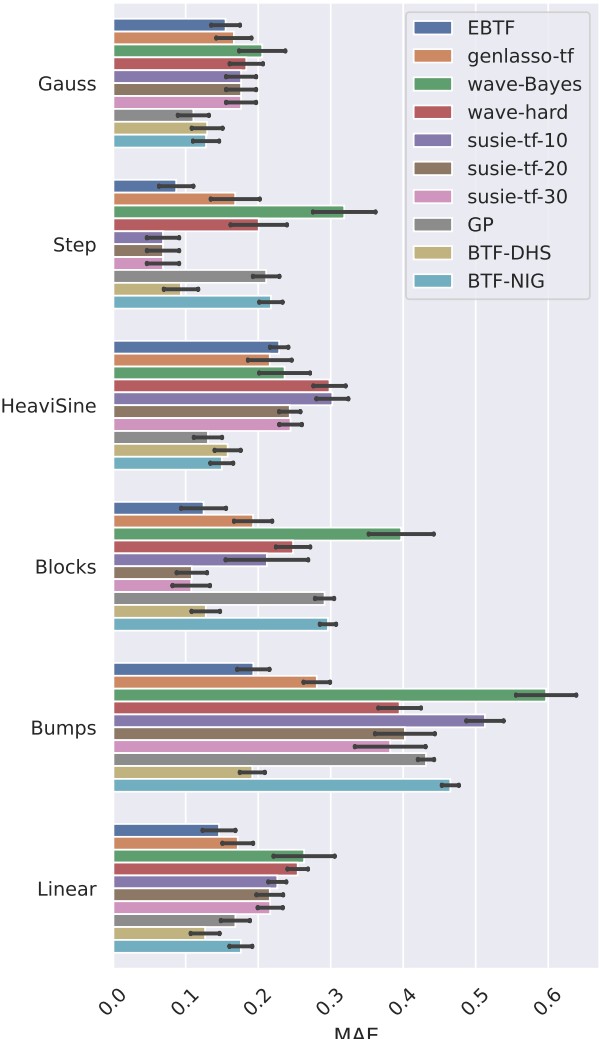

Figure 9: Simulation results: MAE scores of the competing methods across six signal functions.

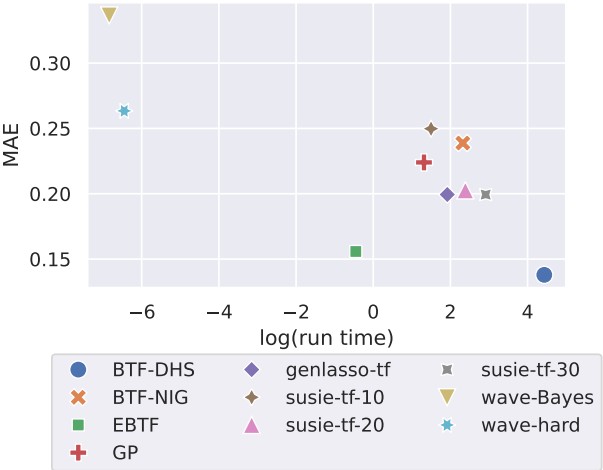

Figure 10: Simulation results: plots of log-transformed runtime vs MAE.

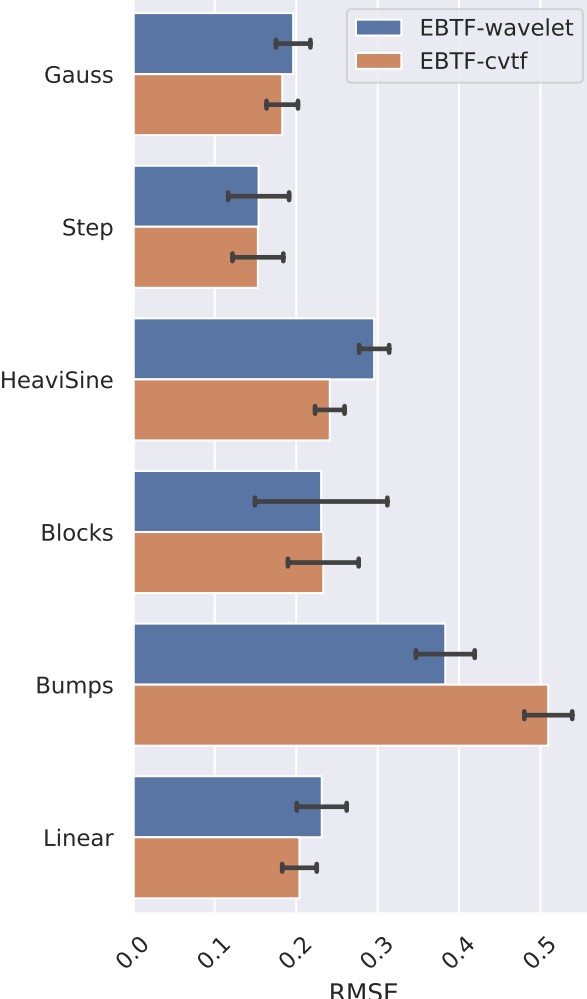

Figure 11: Simulation results: Comparison of the two posterior mean initialization methods for EBTF (wavelet and cross-validated trend filtering). Both methods yield similar performance, but we prefer the wavelet initialization due to its faster computation.

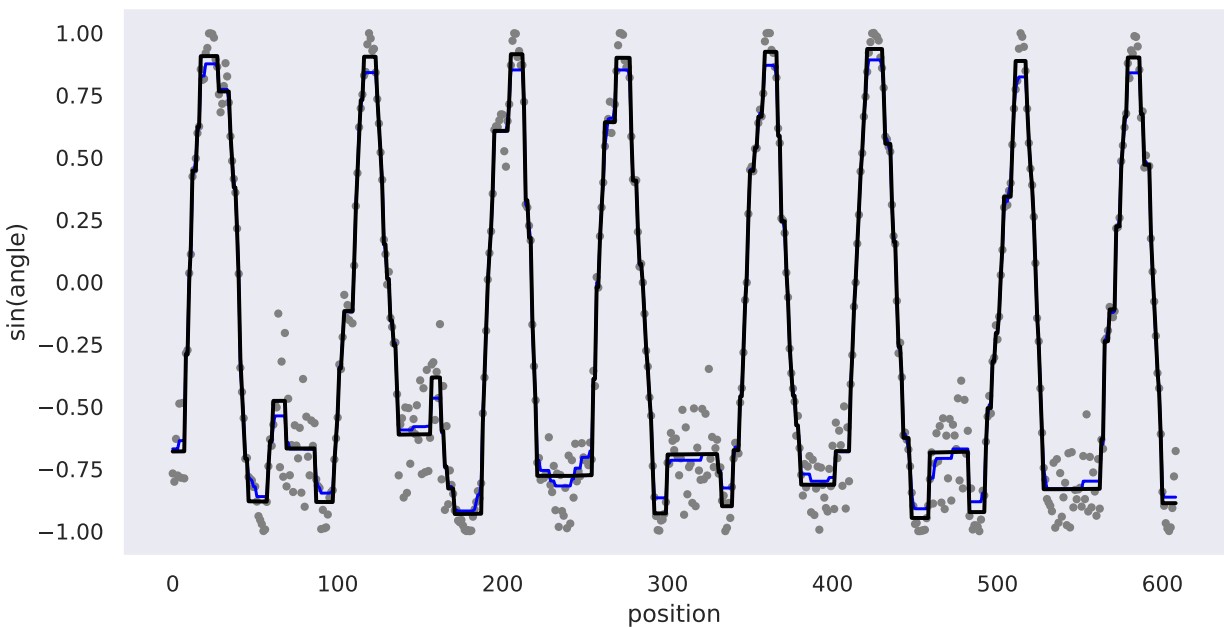

Figure 12: Honey bee movement states. The $x$-axis is the position and $y$-axis is sine of the head angle of a single bee. Black line is the EBTF fitted signal, and blue line is the genlasso-tf estimated signal.

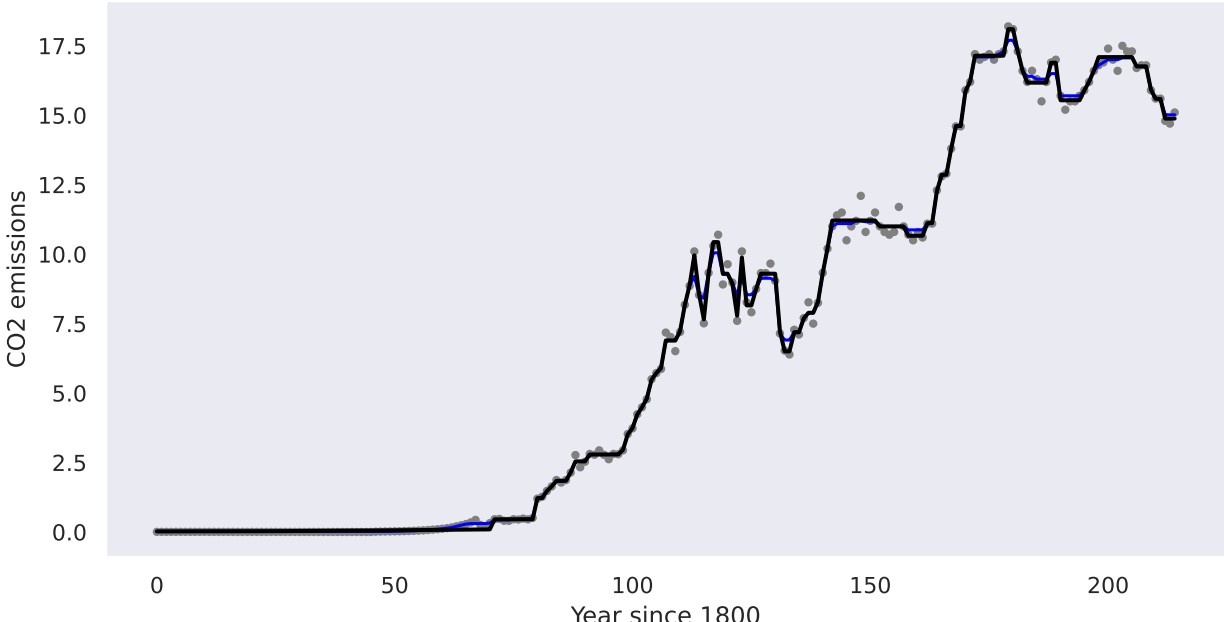

Figure 13: $CO_2$ emissions per person in Canada. Black line is the EBTF fitted signal, and blue line is the genlasso-tf estimated signal.

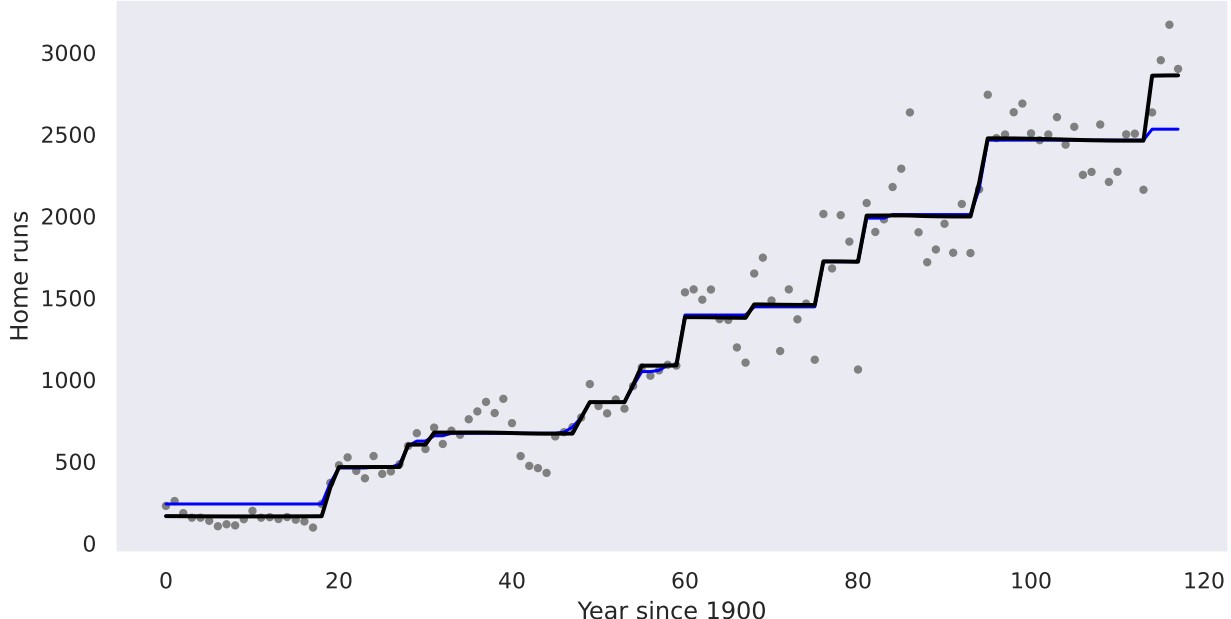

Figure 14: Number of home runs in the American League of baseball since 1900. Black line is the EBTF fitted signal, and blue line is the genlasso-tf estimated signal.

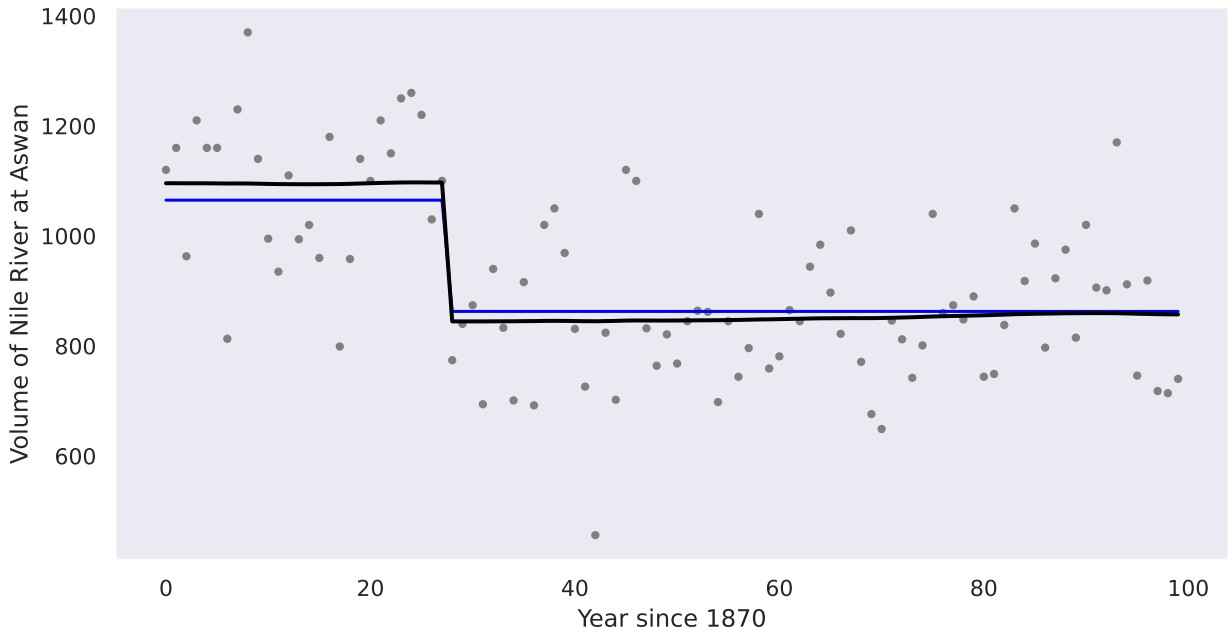

Figure 15: The volume of the Nile river at Aswan over each year. There is a clear change point in 1898 due to a built dam.

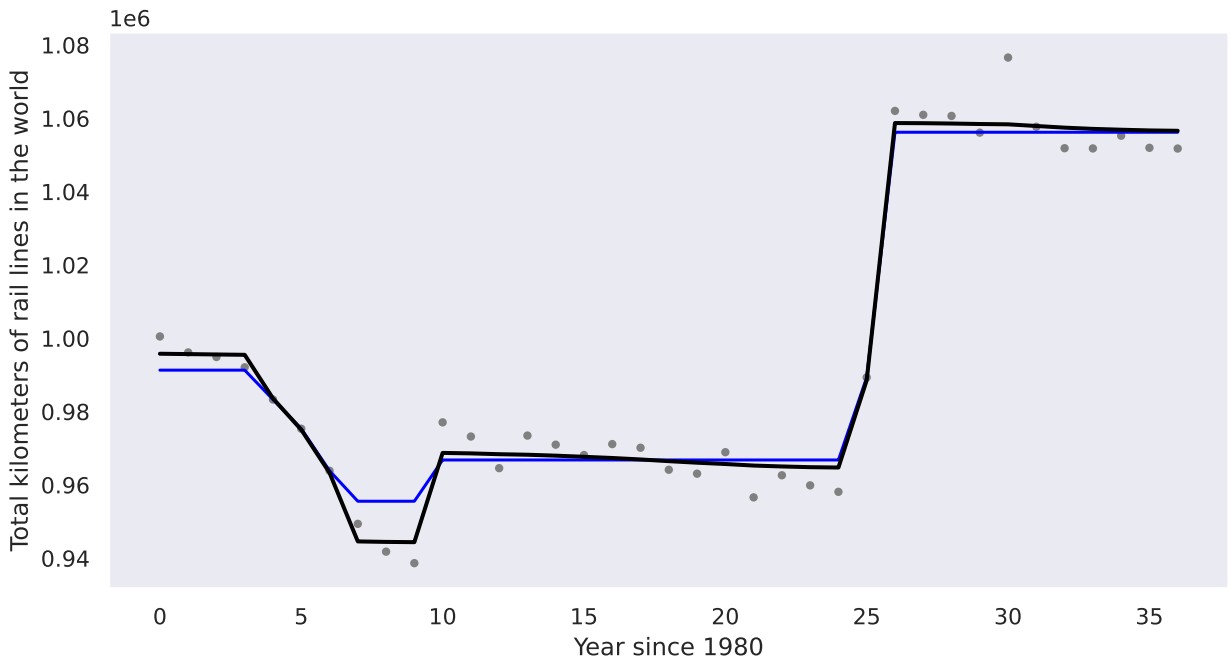

Figure 16: Total length of rail lines in the world, in kilometers.

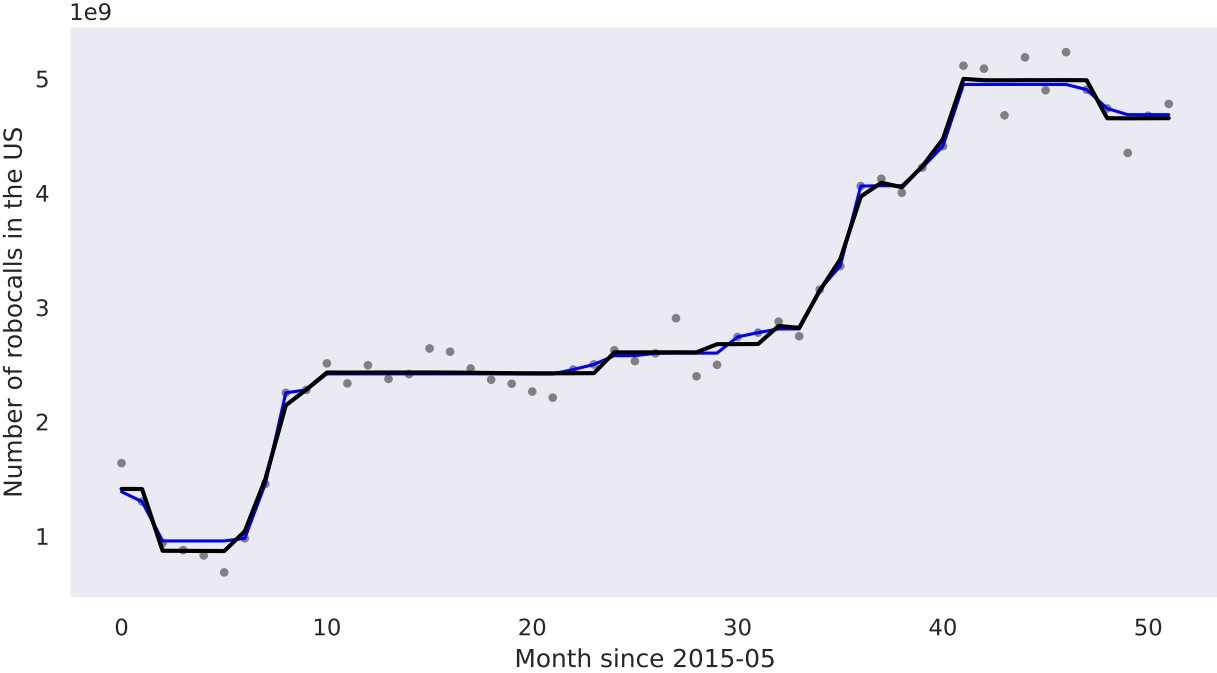

Figure 17: The number of robot calls in US.

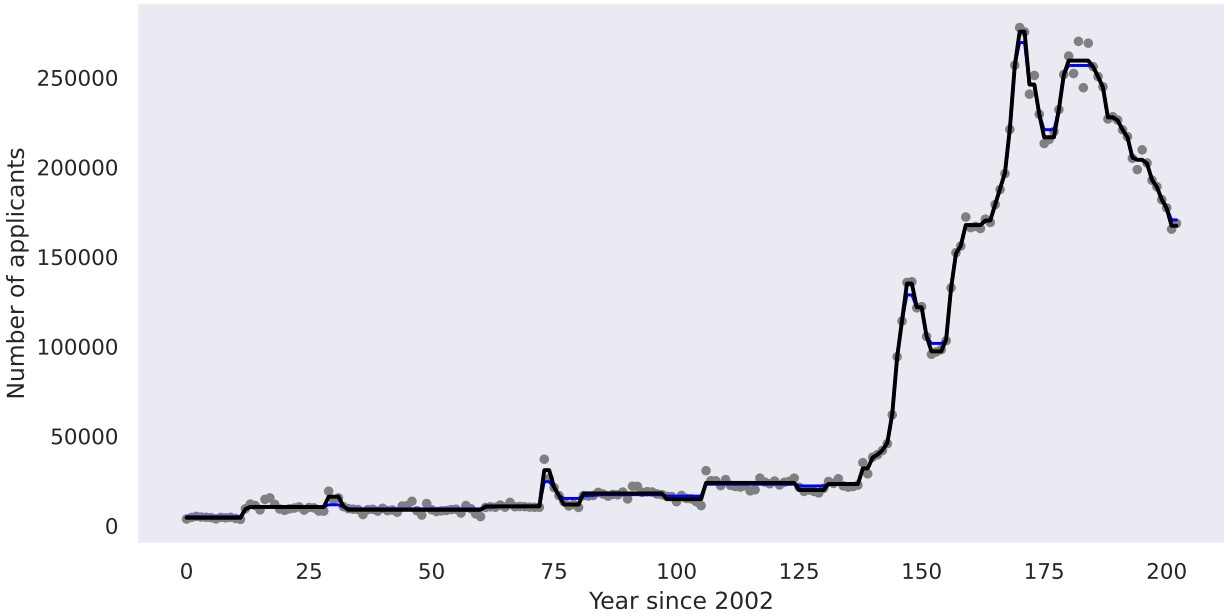

Figure 18: The number of license plate applications in Shanghai since 2002. Two outliers were removed.

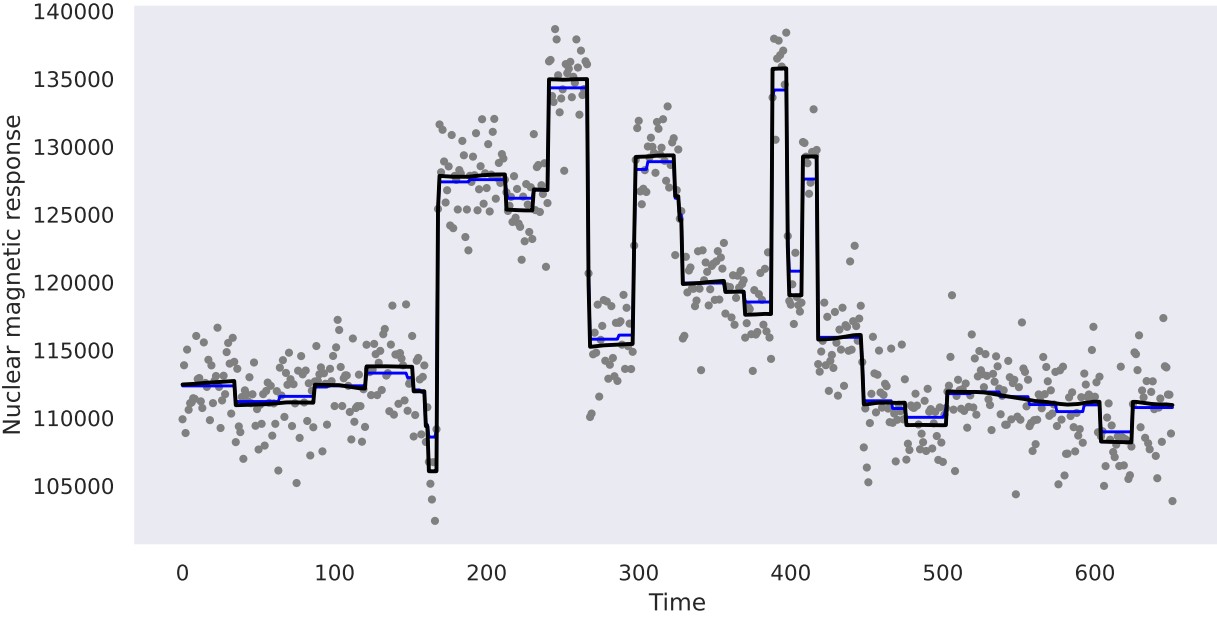

Figure 19: Well-log data. It captures the nuclear magnetic responses over time. The length of the series has been reduced by sampling every 6 time points. Outliers were removed.

## B   Empirical Bayes Gaussian Variance

We give details on the general Gaussian variance problem.

Consider the following model on Gaussian variance: for $i = 1, 2, ..., n$,

$$
\begin{aligned}
x_i | w_i &\sim N(0, \sigma^2 w_i), \\
w_i &\sim \tilde{g}(\cdot),
\end{aligned}
\tag{14}
$$

where $\sigma^2$ is known. An empirical Bayes Gaussian variance procedure returns $\hat{\tilde{g}}$ by maximizing the marginal log likelihood $\sum_i \log p(x_i)$, and calculates the posterior $q_{w_i}(\cdot) = p(w_i | x_i, \hat{\tilde{g}})$. The objective function of EBGV problem is

$$
F_{EBGV} = \sum_i \mathbb{E}_q \log N(x_i; 0, \sigma^2 w_i) + \sum_i \mathbb{E}_q \log \frac{\tilde{g}(w_i)}{q_{w_i}(w_i)}.
$$

The procedure defines a mapping from observations to the estimated prior and posterior distribution, and is denoted as

$$
(\hat{g}, q_{\boldsymbol{w}}) = \text{EBGV}(\boldsymbol{x}, \sigma^2).
$$

In the EBGV problem equation 14, if $w_i$ has a discrete prior as

$$
w_i \sim \text{Discrete}(\sigma_1^2, ..., \sigma_k^2; \boldsymbol{\pi}),
$$

then the marginal distribution of $x_i$ is

$$
p(x_i) = \sum_w p(x_i | w) p(w) = \sum_k \pi_k N(x_i; 0, \sigma^2 \sigma_k^2).
$$

The posterior distribution of $w_i$ is

$$
p(w_i | x_i) = \prod_k \phi_{ik}^{I(w_i = \sigma_k^2)},
$$

where

$$
\phi_{ik} = \frac{\pi_k N(x_i; 0, \sigma^2 \sigma_k^2)}{\sum_k \pi_k N(x_i; 0, \sigma^2 \sigma_k^2)}.
$$

And we have

$$
\mathbb{E}(w_i^{-1} | x_i) = \sum_k \phi_{ik} / \sigma_k^2.
$$

The marginal and posterior distribution are given by their definitions, and the Bayes formula. The expectation of $1/w_i$ follows directly from the definition of discrete random variables.

## C   Algorithms

---

**Algorithm 2** VEB algorithm for fitting EBTF 8 (outline only)

---

**Input:** Data $y_i$, variances $s_i^2$, for $i = 1, 2, ..., n$.
**Init:** Posterior mean $\bar{\boldsymbol{\beta}}$, posterior precision matrix diagonal $\boldsymbol{d}$, and super-diagonal $\boldsymbol{e}$, residual variance $\sigma^2$.
**repeat**
    1. Update $\tilde{g}(\cdot)$ and $q_W$ by solving the EBGV problem equation 16;
    2. Update $q_{\boldsymbol{\beta}}$ by updating the posterior precision matrix (its diagonal and super-diagonal elements only); then $\bar{\boldsymbol{\beta}}$ by solving a (tridiagonal) banded linear system;
    3. Update residual variance $\sigma^2$.
**until** converged

---

## D  Derivations

### D.1  Derivation of VEB updates for fitting model equation 2

Based on the model equation 2, and the posterior distribution equation 6 the evidence lower bound can be written in a vector-matrix form in $\bar{\boldsymbol{\beta}}, V$ as

$$
\begin{aligned}
F_{\text{EBTF}} = & -\frac{n}{2} \log 2\pi\sigma^2 - \frac{1}{2} \sum_{i=1}^{n} \log s_i^2 \\
& -\frac{1}{2\sigma^2}(y^T S^{-1} y - 2 y^T S^{-1} \bar{\boldsymbol{\beta}} + \bar{\boldsymbol{\beta}}^T S^{-1} \bar{\boldsymbol{\beta}} + \text{tr}(S^{-1} V)) \\
& -\frac{1}{2\sigma^2} \bar{\boldsymbol{\beta}}^T D^T W D \bar{\boldsymbol{\beta}} - \frac{1}{2\sigma^2} \text{tr}(D^T W D V) + \frac{1}{2} \log |V| \\
& + \sum_{j=1}^{n-1} \sum_{k=1}^{K} \alpha_{jk} (\log \pi_k - \frac{1}{2} \log 2\pi\sigma^2 \sigma_k^2 - \log \alpha_{jk}),
\end{aligned}
$$

where $S = \text{diag}(s_i^2)$ is the known diagonal variance matrix, $D$ is the first order difference matrix equation 1, and $W = \text{diag}(w_{jj})$ is a diagonal weight matrix, with $w_{jj} = \sum_{k=1}^{K} \alpha_{jk}/\sigma_k^2$.

The update of each parameter (denoted generally as $\theta$) is given by solving the root-finding equation $\partial F / \partial \theta = 0$.

### D.2  Derivation of VEB updates for maximizing the ELBO $F_{\text{MNV}}$

The ELBO for model equation 8 with posterior distribution being equation 9 is

$$
\begin{aligned}
F_{\text{MNV}} &= \mathbb{E}_q \log p(\boldsymbol{y}, \boldsymbol{\beta}, W) - \mathbb{E}_q \log q(\boldsymbol{\beta}, W), \\
&= \mathbb{E}_q \log p(\boldsymbol{y}|\boldsymbol{\beta}) + \mathbb{E}_q \log \frac{p(\boldsymbol{\beta}|W)}{q_{\boldsymbol{\beta}}(\boldsymbol{\beta})} + \mathbb{E}_q \log \frac{g(W)}{q_{\boldsymbol{w}}(W)}.
\end{aligned} \tag{15}
$$

We now show that the variational inference update for $q_{\boldsymbol{\beta}}$ given $q_{\boldsymbol{w}}$ is

$$
\begin{aligned}
\bar{\boldsymbol{\beta}} &= (S^{-1} + D^T \overline{\boldsymbol{W}^{-1}} D)^{-1} \boldsymbol{y}, \\
\boldsymbol{V} &= \sigma^2 (S^{-1} + D^T \overline{\boldsymbol{W}^{-1}} D)^{-1},
\end{aligned}
$$

where

$$
\overline{\boldsymbol{W}^{-1}} = \text{diag}(\overline{w_j^{-1}}),
$$

and given $q_{\boldsymbol{\beta}}$, the update on $\tilde{g}(\cdot), q_{\boldsymbol{w}}$ is obtained by solving an EBGV problem

$$
(\hat{\tilde{g}}, q_{\boldsymbol{w}}) = \text{EBGV}\left( \left( \sqrt{\bar{b}_j^2 + v_{b_j}} \right), \sigma^2 \right), \tag{16}
$$

where

$$\bar{b}_j = (D\bar{\boldsymbol{\beta}})_j,$$
$$v_{b_j} = (DVD^T)_{jj}.$$

Given $q_{\boldsymbol{\beta}}$, the ELBO for updating $q_{\boldsymbol{w}}$ is

$$F_{\mathrm{MNV}}(q_{\boldsymbol{w}}) = \mathbb{E}_{q_w}(\mathbb{E}_{q_{\boldsymbol{\beta}}}(\log p(\boldsymbol{\beta}|W)) + \sum_j \mathbb{E}_{q_w} \log \frac{\tilde{g}(w_j)}{q_{w_j}(w_j)}$$

$$= \sum_j \mathbb{E} \log N(\sqrt{\bar{b}_j^2 + v_{b_j}}; 0, \sigma^2 w_j) + \sum_j \mathbb{E} \log \frac{\tilde{g}(w_j)}{q_{w_j}(w_j)},$$

which is exactly the objective function for EBGV problem. And we have

$$\overline{w_j^{-1}} = \sum_k \phi_{jk}/\sigma_k^2,$$

$$\phi_{jk} = \frac{\pi_k N(\sqrt{\bar{b}_j^2 + v_{b_j}}; 0, \sigma^2 \sigma_k^2)}{\sum_{k'} \pi_{k'} N(\sqrt{\bar{b}_j^2 + v_{b_j}}; 0, \sigma^2 \sigma_{k'}^2)}.$$

Given $q_{\boldsymbol{w}}$, the ELBO related to $q_{\boldsymbol{\beta}}$ is

$$F_{\mathrm{MNV}}(q_{\boldsymbol{\beta}}) = \mathbb{E} \log p(\boldsymbol{y}|\boldsymbol{\beta}) + \mathbb{E}(\mathbb{E}_{q_{\boldsymbol{w}}}(\log p(\boldsymbol{\beta}|W))) - \mathbb{E} \log q_{\boldsymbol{\beta}}(\boldsymbol{\beta}).$$

The update formulas for $\bar{\boldsymbol{\beta}}, V$ are given by solving the root-finding equation $\partial F_{\mathrm{MNV}}(q_{\boldsymbol{\beta}})/\partial \bar{\boldsymbol{\beta}} = 0$, and $\partial F_{\mathrm{MNV}}(q_{\boldsymbol{\beta}})/\partial V = 0$.

Since the marginal distribution in EBGV problem is the Gaussian mixture distribution, the update of prior parameters $\boldsymbol{\pi}, (\sigma_k^2)$ are the same as the ones in VEB updates for fitting model equation 2, with $\phi_{jk} = \alpha_{jk}$. It is obvious the update for $\sigma^2$ is also the same since the ELBO related to $\sigma^2$ in $F_{\mathrm{MNV}}$ is the same as $F_{\mathrm{EBTF}}(\sigma^2)$. Furthermore, the variational inference algorithm is the same as Algorithm 1.

### D.3 Derivation of VEB updates for maximizing the ELBO $F_{\mathsf{MLR}}$

The ELBO $F_{\mathrm{MLR}}$ is

$$
\begin{aligned}
F_{\mathrm{MLR}} =& \mathbb{E}_q \log p(\boldsymbol{y}, \tilde{\boldsymbol{b}}, \boldsymbol{z}) - \mathbb{E}_q \log q(\tilde{\boldsymbol{b}}, \boldsymbol{z}) \\
=& -\frac{n}{2} \log \sigma^2 - \frac{1}{2} \sum_i \log s_i^2 \\
& -\frac{1}{2\sigma^2}(y^T S^{-1} y - 2y^T H \bar{\tilde{\boldsymbol{b}}} + \bar{\tilde{\boldsymbol{b}}}^T H^T S^{-1} H \bar{\tilde{\boldsymbol{b}}} + \mathrm{tr}(H^T S^{-1} H V_{\tilde{b}}) \\
& + \sum_{j,k} \alpha_{jk}(\log \pi_k - \frac{1}{2} \log \sigma^2 \sigma_k^2 - \frac{1}{2\sigma^2 \sigma_k^2}(\bar{b}_j^2 + v_{b_j})) \\
& + \frac{1}{2} \log |V_{\tilde{b}}| - \sum_{j,k} \alpha_{jk} \log \alpha_{jk}, \\
=& -\frac{n}{2} \log \sigma^2 - \frac{1}{2} \sum_i \log s_i^2 \\
& -\frac{1}{2\sigma^2}(y^T S^{-1} y - 2y^T H \bar{\tilde{\boldsymbol{b}}} + \bar{\tilde{\boldsymbol{b}}}^T H^T S^{-1} H \bar{\tilde{\boldsymbol{b}}} + \mathrm{tr}(H^T S^{-1} H V_{\tilde{b}}) \\
& -\frac{1}{2\sigma^2}(\bar{\boldsymbol{b}}^T W \bar{\boldsymbol{b}} + \mathrm{tr}(W V_{\tilde{b}})) + \frac{1}{2} \log |V_{\tilde{b}}| \\
& + \sum_{j,k} \alpha_{jk}(\log \pi_k - \frac{1}{2} \log \sigma^2 \sigma_k^2 - \log \alpha_{jk}),
\end{aligned}
$$

where $W = \text{diag}(w_j)$ and $w_j = \sum_k \alpha_{jk}/\sigma_k^2$.

Let

$$\bar{\tilde{b}} = \begin{pmatrix} \bar{\beta}_1 \\ \bar{b} \end{pmatrix}, V_{\tilde{b}} = \begin{pmatrix} v_{\beta_1} & V_{\beta_1, b} \\ V_{b, \beta_1} & V_b \end{pmatrix}.$$

Given $q_{\tilde{b}}(\tilde{b}) = N(\tilde{b}; \bar{\tilde{b}}, V_{\tilde{b}})$, the inducing posterior distribution on $\beta$ is also multivariate normal, denoted as $q_\beta(\cdot) = N(\beta; \bar{\beta}, V_\beta)$, where $\bar{\beta} = H\bar{\tilde{b}}$ and $V_\beta = HV_{\tilde{b}}H^T$. Based on the relationship between $\beta, b, \tilde{b}$, we also have

$$\bar{b} = D\bar{\beta},$$
$$V_b = DV_\beta D^T.$$

Since $H$ is a triangular matrix with diagonal elements all being 1, we have $|H| = 1, |H^T| = 1$, and $|V_\beta| = |HV_{\tilde{b}}H^T| = |H||V_{\tilde{b}}||H^T| = |V_{\tilde{b}}|$.

Given the above equivalence, the ELBO can be written as

$$\begin{aligned}
F_{\text{MLR}} = &-\frac{n}{2}\log 2\pi\sigma^2 - \frac{1}{2}\sum_{i=1}^n \log s_i^2 \\
&- \frac{1}{2\sigma^2}(y^T S^{-1} y - 2y^T S^{-1}\bar{\beta} + \bar{\beta}^T S^{-1}\bar{\beta} + \text{tr}(S^{-1}V_\beta)) \\
&- \frac{1}{2\sigma^2}\bar{\beta}^T D^T W D\bar{\beta} - \frac{1}{2\sigma^2}\text{tr}(D^T W DV_\beta) + \frac{1}{2}\log|V_\beta| \\
&+ \sum_{j=1}^{n-1}\sum_{k=1}^K \alpha_{jk}(\log \pi_k - \frac{1}{2}\log 2\pi\sigma^2\sigma_k^2 - \log \alpha_{jk}),
\end{aligned}$$

which is exactly the same as $F_{\text{EBTF}}$.

### D.4 Update formulas for sparse EBTF

Let $W_0 = \text{diag}(\alpha_{i0}/\sigma_0^2)$, the updates for empirical Bayes sparse trend filtering are as follows:

1. The update for posterior probabilities are

$$\log \alpha_{ik} \propto \log \pi_k - \frac{1}{2}\log 2\pi\sigma^2\sigma_k^2 - \frac{(D\bar{\beta})_i^2 + (DVD^T)_{ii}}{2\sigma^2\sigma_k^2},$$

$$\log \alpha_{i0} \propto \log \pi_0 - \frac{1}{2}\log 2\pi\sigma^2\sigma_0^2 - \frac{\bar{\beta}_i^2 + V_{ii}}{2\sigma^2\sigma_0^2},$$

$$\log \alpha_{11} \propto \log(1 - \pi_0),$$

then

$$\alpha_{10}, \alpha_{11} \leftarrow \frac{\alpha_{10}}{\alpha_{10} + \alpha_{11}}, \frac{\alpha_{11}}{\alpha_{10} + \alpha_{11}},$$

and

$$\alpha_{ik} \leftarrow \frac{\alpha_{ik}}{\sum_l \alpha_{il}},$$

for $i = 2, 3, ..., n$ and $k = 0, 1, ..., K$.

2. The update for the posterior covariance matrix $V$ and the posterior mean $\bar{\beta}$ are

$$V = \sigma^2(S^{-1} + D^T W D + W_0)^{-1},$$
$$\bar{\beta} = Vy/\sigma^2.$$

3. The update for prior variance $\sigma_k^2$ for $k = 1, 2, ..., K$ and prior probabilities $\boldsymbol{\pi}$ are

$$\sigma_k^2 = \frac{\sum_i (\alpha_{ik}((D\bar{\boldsymbol{\beta}})_i^2 + (DVD^T)_{ii}))}{\sigma^2 \sum_i \alpha_{ik}},$$

$$\pi_0 \leftarrow \sum_{i=1}^{n} \alpha_{i0},$$

$$\pi_k \leftarrow \sum_{i=2}^{n} \alpha_{ik}, \text{ for } k = 1, ..., K,$$

$$\pi_k \leftarrow \frac{\pi_k}{\sum_{l=1}^{K} \pi_l}, \text{ for } k = 0, 1, ..., K.$$

4. Update $\sigma^2$ as

$$\Omega = S^{-1} + D^T WD + W_0,$$

$$\sigma^2 = \frac{y^T S^{-1} y - 2y^T S^{-1} \bar{\boldsymbol{\beta}} + \bar{\boldsymbol{\beta}}^T \Omega \bar{\boldsymbol{\beta}} + \text{tr}(\Omega V)}{2n - 1 + \alpha_{10}}$$

The updates of each parameter (denoted generally as $\theta$) is obtained by solving the root-finding equation

$$\frac{\partial F}{\partial \theta} = 0,$$

where $F$ is the ELBO for EBSTF, defined as

$$\begin{aligned}
F =& \mathbb{E} \log p(\boldsymbol{y}, \boldsymbol{\beta}, \boldsymbol{z}) - \mathbb{E} q_\beta(\boldsymbol{\beta}) q_z(\boldsymbol{z}) \\
=& -\frac{n}{2} \log 2\pi\sigma^2 - \sum_i \frac{1}{2} \log s_i^2 \\
& - \frac{1}{2\sigma^2} (y^T S^{-1} y - 2y^T S^{-1} \bar{\boldsymbol{\beta}} + \bar{\boldsymbol{\beta}}^T S^{-1} \bar{\boldsymbol{\beta}} + \text{tr}(S^{-1} V)) \\
& + \alpha_{11}(\log(1 - \pi_0)) + \sum_{i=1} \alpha_{i0}(\log \pi_0 + \mathbb{E} \log N(\beta_i; 0, \sigma^2 \sigma_0^2)) \\
& - \frac{1}{2\sigma^2} \bar{\boldsymbol{\beta}}^T D^T WD\bar{\boldsymbol{\beta}} - \frac{1}{2\sigma^2} \text{tr}(D^T WDV) + \frac{1}{2} \log |V| \\
& + \sum_{i=2, k=1} \alpha_{ik}(\log \pi_k - \frac{1}{2} \log 2\pi\sigma^2 \sigma_k^2 - \log \alpha_{ik}) - \alpha_{11} \log \alpha_{11}.
\end{aligned}$$

