# OpenReview forum: "Empirical Bayes Trend Filtering Through a Variational Inference Framework"
_TMLR — Accepted by TMLR_

### Review · Reviewer_Qhzt · 2025-01-25

**Summary Of Contributions:**

The authors consider a Bayesian approach to "trend filtering" which is smoothing of a one-dimensional trajectory. They construct a model with an ash-esque prior on the signal change between time-points, and develop a coordinate ascent variational inference procedure for fitting this model. They show promising performance, both computationally and statistically, on simulated data. They demonstrate the method on a small "real world" dataset.

**Audience:**

Yes

**Claims And Evidence:**

Yes

**Requested Changes:**

For the real data hold out some time points from the model and ask how well they can be imputed. I would like to see at least two more datasets with different characteristics - the motorcycle data looks pretty smooth, so having data with substantial jumps would be a nice addition. Ideally additionally having some larger datasets (financial time series maybe?) would be valuable. Especially since the VI is motivated by scalability it is disappointing to only see an application to a tiny dataset.

**Strengths And Weaknesses:**

The paper is clearly written. The only slight oddness to me in the presentation was the sparse version being described at the end after the results for the non-sparse version. It would be cleaner to have the sparse EBTF described along with the main approach and then added as a comparison method in Table 1 - presumably it should do well on the less smooth simulated data. As an additional point here: the sparse EBTF doesn't really feel like a different model, since as far as I can see it is equivalent to fixing one of the variances in EBTF to a small value.

The modeling and inference is all very reasonable. The simulated results are also fine. The "real data" feels very limited: this is a small dataset and there is no quantitative assessment, which seems relatively straightforward in this context: see "Requested Changes".

Is there a reason smoothing approaches like Gaussian process regression aren't compared to? They would be a poor fit for data with jumps (although there are GP change point models) but would feel natural for something as smooth as the motorcycle data.

Minor comments:

The authors say in the intro they will denote the vector beta as (beta_i) and W as diag(w_i) which is poor notation imo. i refers to one element. If you want to specify all the individual elements simply use beta, or specify {beta_i:i=1...n} or similar. Fortunately they don't actually seem to use this notation despite introducing it.

What does the cdot between D^(1) and D^(k) denote? I believe this is matrix multiplication, in which case there should be no symbol there: matrix multiplication is implied. If it is Hadamard that should be specified.

I'm not aware of people calling what is being done here "variational empirical Bayes" (VEB), I just call this variational Bayesian EM (VBEM), where the M steps are on the optimized parameters. I don't hate VEB though.

sigma and pi are parameters, _not_ variational parameters (page 4)

Calling the updates a "Theorem" is overselling. A "computation" maybe?

Table 1 would be easier to grok as a barchart, and R^2 is easier to interpret than RMSE.

---

> ### Author Response · Authors · 2025-02-23
>
> Thank you for your review and valuable feedback!  Please find our response to the comments below:
>
> ## Strength and Weakness
>
> >"The only slight oddness to me in the presentation was the sparse version being described at the end..."
>
> The primary focus of our paper is to introduce the general empirical Bayes formulation of trend filtering and explore different modeling strategies. We present sparse EBTF after the main modeling sections to highlight its flexibility as an extension of the dynamic linear model framework, contrasting it with other formulations. While sparse EBTF is indeed a special case of EBTF rather than an entirely new model, its sparsity-inducing prior alters the posterior behavior and demonstrates the advantages of combining a dynamic linear model with empirical Bayes estimation.
>
> >"Is there a reason smoothing approaches like Gaussian process regression aren't compared to..."
>
> The main reason we did not initially include GPs in our comparison is that GPs are not locally adaptive, so they do not adapt to varying smoothness levels as effectively as trend filtering or wavelet-based methods. We acknowledge that GPs can be useful for globally smooth signals like the motorcycle data. In our revised version, we have now included GPs as a method in all simulation and real-data examples to provide a more comprehensive comparison.
>
> ## Minor comments
>
> 1. Thank you for pointing out the notation issue with $\beta$ and $W$. We have corrected it by explicitly specifying $i = 1,2,\dots,n$  and used the proper vector notation in equations (3) and (8).
> 2. The dot denotes matrix multiplication. We have removed it for clarity since matrix multiplication is implied.
> 3. We introduce "VEB" to emphasize the empirical Bayes perspective due to the learning of priors, but we acknowledge that the algorithm aligns with variational EM and have clarified this in the text.
> 4. Corrected.
> 5.  We have replaced "Theorem" with "VEB Updates" to better reflect the nature of the results.
> 6. We appreciate the suggestions. We have revised the numerical results by adding $R^2$ as a metric and replacing Table 1 with a bar chart for better visualization.
>
> ## Requested Changes
>
> We have added three additional datasets (ETTh1, illness, and weather) that feature longer sequences (ranging from $10^3$ to $10^4$) and varying levels of smoothness, including substantial jumps. To better assess predictive performance, we have also conducted holdout experiments on these datasets. These additions provide a more comprehensive evaluation of EBTF's scalability and adaptability. Please refer to section 4.2 for details in the revised version.

---

### Review · Reviewer_tkuc · 2025-01-29

**Summary Of Contributions:**

This paper introduces a variational empirical Bayes (VEB) approach to trend filtering. In particular, a variational solution to a first-order trend filtering problem is presented, using shrinkage priors based on mixtures of normals. The problem is originally formulated in terms of a dynamic linear model and a coordinate ascent algorithm presented; it is later shown that various other formulations of trend filtering lead to the same parameter updates. Experiments on toy datasets show strong quantitative results with faster runtime than similarly performing methods.

**Audience:**

No

**Claims And Evidence:**

No

**Requested Changes:**

## Critical
- Please address the above concerns regarding the experiments.
- Nearly all citations should be using `\citep`, but are not.
- Please include standard errors for all results in Table 1, even if the general scale is similar to the first three datasets.
- Several papers cited reference the arXiv version of a paper, even though a more recent journal or conference version is available (e.g., Ning & Ning (2024), Castillo & Roquain (2020), and Kim et al. (2024)). If there is no particular reason to cite the arXiv version, I suggest the authors consider citing the latest version. Note that I did not check all references for this, and there may be more examples.

## Minor
- There are many places in the bibliography with missing capitalization, either in proper nouns or in journal/publisher titles; for example, "Pareto", "Cambridge University Press", and "Signal Processing". "\ell_1" should also use math mode.
- At the beginning of 2.3.2, it should read "for the prior *in equation* 3 [...]".
- The acronym "MLR" (multiple linear regression) is not introduced.
- The trend $\beta$ is not introduced in Section 2.1.

# References

Kim, Y., Wang, W., Carbonetto, P., & Stephens, M. (2024). A flexible empirical Bayes approach to multiple linear regression and connections with penalized regression. _Journal of Machine Learning Research_, _25_(185), 1-59.

Ning, Y. C. B., & Ning, N. (2024). Spike and slab Bayesian sparse principal component analysis. _Statistics and Computing_, _34_(3), 118.

Carvalho, C. M., Polson, N. G., & Scott, J. G. (2009). Handling sparsity via the horseshoe. In _Artificial Intelligence and Statistics_ (pp. 73-80). PMLR.

Castillo, I., & Roquain, É. (2020). On Spike and Slab Empirical Bayes Multiple Testing. _The Annals of Statistics_, _48_(5), 2548-2574.

Jaakkola, T. S., & Jordan, M. I. (1997, January). A variational approach to Bayesian logistic regression models and their extensions. In _Sixth International Workshop on Artificial Intelligence and Statistics_ (pp. 283-294). PMLR.

Seeger, M., & Bouchard, G. (2012). Fast variational Bayesian inference for non-conjugate matrix factorization models. In _Artificial Intelligence and Statistics_ (pp. 1012-1018). PMLR.

Quiroz, M., Kohn, R., Villani, M., & Tran, M. N. (2019). Speeding up MCMC by efficient data subsampling. _Journal of the American Statistical Association_.

**Strengths And Weaknesses:**

## Strengths
- The paper is generally well-written, and the theoretical results are sound.
- On the datasets evaluated, the proposed method is indeed fast and accurate relative to the baselines. The empirical results suggest that EBTF underfits less than competing methods.
- The sparse EBTF extension is a nice way to deal with periodically dropped signals, and shows the flexibility of the dynamic linear model formulation of trend filtering.

## Weaknesses

**Regarding Audience**
I'm somewhat skeptical of the audience; personally, I don't feel that trend filtering is a common interest in the machine learning community, and it is more typically discussed in statistics and signal processing communities. While variational inference may be more generally of interest, the coordinate ascent algorithm derived in this work is a relatively straightforward application of the VEB framework.

**`susie-tf-20` and Figure 1**
I find Figure 1 a bit misleading. `susie-tf-20` has an outlier in performance on one dataset, which brings its average RMSE up dramatically. Aside from datasets with pathologically "spiky" data, `susie-tf` seems quite strong.  Plus, averaging RMSEs like this (ignoring the scale of different tasks) is not ideal in the first place.

**Code Availability**
The paper mentions that "all the code and analysis are available in the package `ebtfPy` on GitHub." I did not search for this package as I was concerned it might break the authors' anonymity, and I did not see any anonymized version provided, so I could not look at the code. I would be interested to look at the code, if an anonymized version can be provided.

**On the Use of MCMC**
It's mentioned that "while it's possible to use MCMC for sampling from the posterior, it's intractable for large-scale datasets." This is true, of course, but I don't think it is a major concern for datasets of the size considered in this paper ($n = 1024$ in the experiments), at least not to the level of "intractability." For example, the listed reference (Quiroz et al., 2019) uses datasets several orders of magnitude larger in their experiments. I would thus be interested to see a comparison in run times between MCMC and VB.

**On the Extent of Experiments**
I find the experiments somewhat lacking. As an example, the datasets with quantitative results are somewhat simplistic. and finding more realistic datasets where qualitative performance may be evaluated would significantly strengthen the results in my opinion. As another example, using ash priors and variational Bayes introduces two moving parts that are not properly ablated: how much performance in terms of accuracy is lost by resorting to variational inference, and how much slower is MCMC? There is also mention that the theory in the paper holds for higher-order trend filtering problems, but there are no experiments. Finally, ash priors are used somewhat arbitrarily; it would be nice to investigate the effect of different shrinkage priors in trend filtering.

## Questions
- In the simulations, how is $\hat{\beta}$ chosen for EBTF? As the mean of the variational posterior?
- The initializations in Remark 2.2 seem reasonable, but how much do they matter? For example, if $\beta_i$ is intiialized as $y_i$, are results significantly worse? What if they are initialized using the results of `susie-tf-20` or `genlasso-tf`?
- Kim et al. (2024) propose a VEB approach to multiple linear regression. Given the equivalence presented in Section 3.2, could the authors further comment on the relevance of Kim et al. (2024) to EBTF? Are the main algorithmic differences using banded solvers?
- Why are $L=10$ and $L=20$ chosen for `susie-tf`?

---

> ### Author Response · Authors · 2025-02-23
>
> Thank you for your review and insightful feedback. Below are our responses to your comments.
>
> ## Weakness
>
> 1. Trend filtering is relevant for time-series forecasting and structured sparsity modeling within machine learning. Our work enhances the scalability of Bayesian trend filtering by leveraging a variational empirical Bayes (VEB) approach, where we unify multiple formulations under a general EB-VI framework. Beyond trend filtering, our methodology contributes to the broader machine learning field by demonstrating how empirical Bayes can enhance variational inference for structured regression problems. Additionally, we extend our method to sparse EBTF, and show its flexibility in applying these techniques to a wider range of structured prediction and adaptive smoothing tasks.
> 2. We have updated the presentation of simulation results and replaced Table 1 with a bar chart for improved clarity. In real-data experiments, to ensure RMSE values are comparable across datasets, we now normalize RMSE by dividing it by the RMSE of a mean predictor. Figure 2 (previously Figure 1) focuses on the runtime of different methods, complementing Figure 1, which presents detailed RMSE results for each dataset. In simulation susie-tf indeed performs well in almost all dataset except bumps.
> 3. We have included the code as ebtfPy.zip in the Supplementary Material.
> 4. Thank you for the suggestion. We have added Bayesian trend filtering (BTF) for comparison in the simulations. As shown in Figure 2, MCMC-based methods generally run significantly longer than VI-based methods.
> 5. Discussed in the requested changes below.
>
> ## Questions
>
> 1. Yes, we use the posterior mean of the variational distribution.
> 2. Initialization choices do impact performance, and we choose the wavelet method because it is very fast. If $\beta$ is initialized at $y$, it typically does not converge, and the learned curve gets stuck at $y$. In our experience, we experimented with initializing at genlasso-tf, and the results were generally similar to wavelet initialization. We did not use susie-tf due to runtime considerations.
> 3. We highlight the connections between EBTF and the multiple linear regression (MLR) approach. As discussed in Section 3.2, EBTF can be reformulated as an MLR problem with a sparsity-inducing prior and an appropriately constructed design matrix. However, we argue that the EBTF formulation is more flexible, allowing extensions such as sparse EBTF, which are less straightforward in an MLR framework.
> 4. The default choice in susie is $L=10$ and for completeness, we tested $L=20$. However, increasing $L$ further slows down the fitting without significant performance gains in the simulation. In simulations, we tested and added $L=30$  and found that it did not improve much compared to $L=20$. In the new real data examples, we used $L=500$ to capture more trends in susie-tf.
>
> ## Requested Changes
> ### Critical
>
> 1. Experiments:
>
>    (1). We have added MCMC in the simulations and included more real-world datasets in the real-data section.
>
>    (2). While our approach extends naturally to higher-order trend filtering, we focused on 0th-order cases as they already demonstrate the effectiveness of EBTF. A key contribution is the unification of different formulations, which holds for all orders of trend filtering. Additionally, our framework is flexible enough to support variants like sparse EBTF, which would be less straightforward under alternative formulations.
>
>    (3). We chose ash priors due to their flexibility as a generalization of spike-and-slab priors, which are widely used in shrinkage estimation. Exploring continuous shrinkage priors such as horseshoe priors within a variational inference framework is an interesting direction for future work.
> 2. Thanks for pointing out the citation issues—we appreciate it.
>
>    (1). We have corrected the citation format to use `\citep` where appropriate.
>
>    (2). We have updated citations to reference the latest journal or conference versions instead of arXiv when available.
>
> 3. We have replaced Table 1 with a bar chart for improved readability. Standard errors are now displayed on top of each bar to enhance visualization for all methods and all datasets in simulations.
>
> ### Minor
>
> Thanks for all the suggestions! We have addressed all of them in the revision.

---

> > ### Comment · Reviewer_tkuc · 2025-02-28
> > **Response to the Authors [1/2]**
> >
> > Thanks for the revised version of the manuscript. I have a few remaining comments.
> >
> > > We have updated the presentation of simulation results and replaced Table 1 with a bar chart for improved clarity.
> >
> > Thanks! Though, I think the bar chart would be easier to read if grouped by dataset, rather than model. Right now, it is difficult to compare the performance of different models on any given dataset.
> >
> > > We have included the code as ebtfPy.zip in the Supplementary Material.
> >
> > Great, thanks -- the code seems nicely documented and easy to use!
> >
> > Unfortunately, looking through the code, I have some major concerns about the GP experiments on real data. It looks as though the data is never normalized, which is a critical (and computationally trivial) preprocessing step when using GPs.
> >
> > For example, I modified the `GP_sklearn` to normalize the training y values and ran `python realdata/main.py --n 1024 --data_name illness`. The resulting RMSE ratio goes from 0.365 -> 0.137, a pretty dramatic change. Moreover, RBF kernels are not a great choice for data we expect not to be smooth; using the Matern-3/2 kernel instead (which would be the standard choice for these datasets, in my opinion)  provides better results for all real datasets. I've compiled results below:
> >
> > | Dataset | RBF (no normalization) | Matern-3/2 (no normalization) | RBF (normalized) | Matern-3/2 (normalized) |
> > |---------|------------------------:|-------------------------------:|-----------------:|------------------------:|
> > | Illness | 0.365                  | 0.134                         | 0.137            | **0.133**                  |
> > | weather | 0.247                  | 0.187                         | 0.161            | **0.145**                  |
> > | ETTh1   | 0.400                  | 0.229                         | 0.230            | **0.220**                  |
> >
> > The normalized version also runs significantly faster. These together fundamentally change the interpretation of the results in the real data section. I've pasted my changes to the `GP_sklearn` class below, in case I've made an error.
> >
> > > We have added Bayesian trend filtering (BTF) for comparison in the simulations.
> >
> > It is quite surprising to me how poorly BTF performs -- do the authors have any intuitions on why this might be the case? E.g., could this be due to initialization, and would things work better if using the same wavelet initialization as EBTF?
> >
> > Also, part of the reason I suggested an MCMC solution was to properly ablate the benefits from the ash prior and the variational inference -- while the included example shows the speed benefits of variational Bayes over MCMC, it still doesn't properly ablate the benefits of EBVI (in general) and ash priors (more specifically).
> >
> > To be a bit more explicit: EBTF performs much better than BTF in your experiments, but two things change: the prior used, and the inference method. It is thus impossible to tell how much benefit in accuracy is from the prior (dynamic horseshoe -> ash), and how much was due to the inference method (MCMC -> variational EM). In other words, is the disparity between BTF and EBTF in the current results better explained by poor mixing in the MCMC solution, or by better properties in the ash prior (or both)?
> >
> > > Initialization choices do impact performance [...]
> >
> > Thanks for this clarification. I think including these experiments in an appendix (or at least underscoring the practical relevance in Remark 2.1) would be a valuable addition to the manuscript.
> >
> > > The default choice in susie is $L = 10$ [...]
> >
> > Thank you for the clarification!

---

> > > ### Comment · Reviewer_tkuc · 2025-02-28
> > > **Response to Authors [2/2]**
> > >
> > > Below are my changes to the `GP_sklearn` class:
> > >
> > >
> > > ```python
> > > class GP_sklearn:
> > >     def __init__(self, kernel="RBF", normalize_y_train=True):
> > >         self.kernel = kernel
> > >         self.model_name = f"GP_{kernel}"
> > >         self.normalize_y_train = normalize_y_train
> > >
> > >     def fit(self, y):
> > >         """
> > >         Apply Gaussian Process regression to a given array y with specified kernel.
> > >
> > >         Parameters:
> > >         - y: NumPy array containing the input data.
> > >         - kernel: Kernel type for the Gaussian Process (default is 'RBF').
> > >
> > >         Returns:
> > >         - fit_gp: NumPy array containing the Gaussian Process regression output.
> > >         """
> > >         start_time = timeit.default_timer()
> > >         n = len(y)
> > >         X = np.arange(n).reshape(-1, 1)
> > >
> > >         # Normalize inputs
> > >         if self.normalize_y_train:
> > >             y_mu = y.mean()
> > >             y_std = y.std()
> > >         else:
> > >             y_mu = 0.0
> > >             y_std = 1.0
> > >
> > >         y_fit = (y - y_mu) / y_std
> > >
> > >         if self.kernel == "RBF":
> > >             kernel = ConstantKernel(1.0) * RBF(
> > >                 length_scale=1.0, length_scale_bounds=(1e-2, 1e2)
> > >             ) + WhiteKernel(noise_level=1)
> > >         elif self.kernel == "Matern32":
> > >             kernel = ConstantKernel(1.0) * Matern(
> > >                 length_scale=1.0, length_scale_bounds=(1e-2, 1e2)
> > >             ) + WhiteKernel(noise_level=1)
> > >         else:
> > >             raise ValueError("Unsupported kernel type.")
> > >
> > >         gp = GaussianProcessRegressor(kernel=kernel)
> > >         gp.fit(X, y_fit)
> > >         fit_gp = gp.predict(X)
> > >         self.mu = fit_gp.squeeze() * y_std + y_mu
> > >         self.fitted_model = gp
> > >         self.run_time = timeit.default_timer() - start_time
> > > ```

---

> > > ### Author Response · Authors · 2025-03-02
> > >
> > > Thanks for your comments! We have incorporated the suggested revisions into the manuscript. Below are our responses:
> > >
> > > > I think the bar chart would be easier to read if grouped by dataset, rather than model.
> > >
> > > Yeah agreed. In Figure 1, we have grouped the results by dataset to make it easier to compare model performance within each dataset.
> > >
> > > > I have some major concerns about the GP experiments on real data...
> > >
> > > Thank you for pointing out the importance of normalization in GP, the choice of kernel, and for providing the revised code!  We have now incorporated standardization in our experiments and observed improvements in both runtime and predictive performance of GP. We have updated the relevant figures for both the simulation and real data analysss. While the simulation results remain largely unchanged, the real data results show significant improvements on GP—particularly in capturing the illness data trend more effectively (as shown in Figure 5(a)). Additionally, the GP's RMSE on the holdout set is much lower after normalization making it comparable to the other methods. In terms of runtime, we also observed improvements; for instance, the log-runtime for the $n = 4096$ holdout test has decreased from over 4 to below 4 (Figure 6(b)).
> > >
> > > > how poorly BTF performs -- do the authors have any intuitions on why this might be the case?
> > >
> > > Thank you for pointing this out! We were also initially surprised by BTF's poor performance. We investigated the fitted curve from BTF and found that it failed to capture any trend in some simulation functions, and the posterior mean curve is basically flat (a constant function). Upon further examination, we identified that this issue is likely due to the implementation of the `dsp` package when using \( D = 0 \) (zeroth-order trend filtering). We then experimented with setting \( D = 1 \) in the `btf` function and observed significantly improved results. We re-ran the simulations with BTF using \( D = 1 \) and tested two priors: the dynamic Horseshoe prior (DHS) and the normal-inverse-gamma (NIG) prior. With these adjustments, BTF now performs much more competitively, and in fact, the BTF-DHS is the best-performing method in terms of the overall RMSE metrics as shown in Figure 2.
> > >
> > > > It still doesn't properly ablate the benefits of EBVI (in general) and ash priors (more specifically)
> > >
> > > Regarding the prior vs inference method: generally, spike-and-slab priors (ash prior as a subcase) are less suitable for MCMC because they are discrete mixture priors (while DHS and NIG are continuous shrinkage priors), and posterior inference can be computationally demanding with a large number of variables due to the large model space (for example, with the spike and slab components, that's $2^p$ combinations with $p$ variables) (Piironen & Vehtari, 2017). So for spike-and-slab-type priors, variational inference or other optimization-based posterior approximation methods, such as expectation propagation, are typically preferred. Therefore, we do not expect MCMC with the ash prior or in general spike and slab prior to outperform BTF-DHS in terms of predictive accuracy, as BTF-DHS has already demonstrated strong performance in our simulations. Exploring continuous shrinkage priors such as horseshoe priors within a variational inference framework is an interesting direction to us for future work. The advantage of EBTF lies in the combination of empirical Bayes and variational EM, which enables fast curve fitting and optimization-based parameter tuning and posterior calculation while gives strong performance. Additionally, EBTF can serve as an efficient refinement on top of other methods, such as wavelet-based approaches and cross-validated trend filtering, as EBTF uses them for initialization but consistently produces better curve fitting.
> > >
> > > > Including these experiments in an appendix (or at least underscoring the practical relevance in Remark 2.1) would be a valuable addition to the manuscript.
> > >
> > > We have added the comparison of using wavelet and cross-validated trend filtering as initialization on the simulation data in the appendix, Figure 11, showing that the performance difference between the two is not significant. Additionally, we have discussed this in Remark 2.1.
> > >
> > >
> > > ## Reference
> > >
> > > Juho Piironen. Aki Vehtari. "Sparsity information and regularization in the horseshoe and other shrinkage priors." Electron. J. Statist. 11 (2) 5018 - 5051, 2017. https://doi.org/10.1214/17-EJS1337SI

---

### Review · Reviewer_JeQM · 2025-02-13

**Summary Of Contributions:**

This paper introduces a Bayesian trend filtering algorithm based on variational inference. Building on an existing probabilistic model, the authors formulate variational distributions and derive a coordinate ascent algorithm for inference. They further explore alternative parameterizations that yield the same variational objectives, arguing that these alternatives enhance modeling flexibility while preserving analytical tractability. The proposed method is evaluated on both synthetic and real-world datasets, demonstrating improved speed and accuracy compared to baseline approaches.

**Audience:**

Yes

**Broader Impact Concerns:**

I don't see any potential broader impact concerns from this work.

**Claims And Evidence:**

No

**Requested Changes:**

- The paper primarily derives a coordinate ascent variational inference algorithm, but beyond standard calculations such as writing down the ELBO, taking derivatives, and solving for updates, it is unclear what the core technical contribution is. Please clarify whether there are additional methodological innovations that go beyond these routine steps, as this is crucial for assessing the novelty of the work.
- The necessity of the three model variants (the primary model, MNV, and MLR) is not entirely clear. While the paper states that Section 5 is intended to illustrate their differences and motivations, I found it difficult to grasp the key takeaways from that discussion. Could you further elaborate on why these variants are needed, in what scenarios they offer advantages, and how they compare empirically?

**Strengths And Weaknesses:**

Strength:
- The paper is generally well-written and easy to follow.
- The proposed variational Bayes algorithm is more scalable than baseline methods, at least on synthetic data.

Weaknesses:
- The novelty appears limited. The probabilistic formulation of the trend filtering problem is well-established in the literature, and deriving a coordinate ascent algorithm for variational posteriors is relatively straightforward, primarily involving standard calculations rather than substantial technical innovations.
- The real-data experiments lack quantitative evaluation metrics, making it difficult to assess the extent of improvement over baseline methods.
- While the paper introduces multiple variants of the proposed model and discusses their trade-offs, the explanations remain somewhat vague. It is unclear from both the text and the experiments which variants are preferable under specific circumstances, raising questions about their necessity.

---

> ### Author Response · Authors · 2025-02-23
>
> We appreciate your review and constructive feedback. Below, we address your comments in detail.
>
> ## Weakness
>
> 1. Our key contribution lies in reframing trend filtering within a broader empirical Bayes and variational inference framework, and also connect it to multiple existing formulations. While deriving a coordinate ascent algorithm follows standard VI procedures, our approach integrates empirical Bayes learning with structured dependencies, offering a computationally efficient alternative to traditional Bayesian trend filtering methods. Additionally, we extend our framework to sparse EBTF, demonstrating its flexibility beyond standard trend filtering and its potential applicability to a wider range of structured regression problems.
>
> 2. We have included more real dataset examples and incorporated quantitative performance metrics such as RMSE, MAE, and $R^2$ in the simulation and real-data experiments to provide more comparisons among all the methods. Our simulations already demonstrate strong performance and runtime improvements, and these findings are now explicitly validated in real-data settings.
>
> 3. The connection among different formulations is one of our key contribution. The three formulations have different roles and advantages:
>
> - DLM (Primary Model): Most flexible for Bayesian extensions, useful when incorporating structured priors.
> - MNV: Allows modular prior specification, particularly useful when selecting shrinkage priors via empirical Bayes.
> - MLR: Links to classical sparse regression, making it compatible with existing sparse regression solvers.
>
> We illustrate the flexibility of DLM approach by providing a sparse EBTF extension in section 5, which is not straightforward using other two formulations.
>
> ## Requested Changes
>
> 1. While our variational inference  derivation follows standard steps, our key contribution is in unifying multiple formulations of trend filtering within a broader empirical Bayes framework. By bridging empirical Bayes, VI, and structured sparsity, we develop a computationally efficient method that preserves posterior dependencies while scaling to large datasets, which is a challenge in Bayesian trend filtering. Additionally, our sparse EBTF extension expands the applicability of our primary approach beyond standard VI, and shows its flexibility in structured regression and adaptive smoothing problems.
> 2. The three formulations (DLM, MNV, and MLR) have distinct roles in model interpretation and flexibility. The DLM formulation provides a flexible Bayesian framework, the MNV formulation enables modular prior selection via empirical Bayes variance estimation, and the MLR formulation connects directly to sparse regression techniques, linking our work to broader ML literature. Empirically, each variant has advantages: MNV improves modularity and interpretability for hierarchical shrinkage, MLR aligns with existing sparse regression solvers, and DLM supports structured prior extensions such as the sparse EBTF.

---

### Author Response · Authors · 2025-02-23

We appreciate all the valuable feedback and suggestions! We have carefully addressed the comments and incorporated the necessary revisions in our updated submission.

---

### Decision · Action_Editor_PDti · 2025-03-19

**Recommendation:** Accept with minor revision

**Comment:**

See comments above.

One requirement I will impose prior to publication is to turn all theorem/lemma/etc statements into just regular mathematics interspersed with text both in the main text and appendix (appropriately formatted without an environment). These are all overclaimed, as they are all just routine derivations.

**Audience:**

The main consideration during the reviewer discussion was regarding whether the paper meets the bar of being of interest to a nontrivial subset of the TMLR community. The paper proposes a very routine use of variational Bayes on a known model, makes some connections to a few other formulations of that model, and presents some empirical results that demonstrate modest improvements on some problems versus existing techniques. The reviewers were generally on the fence about this, as I am myself. But given that the reviewers all agreed that the paper is of reasonable quality overall, I am inclined to accept.

**Claims And Evidence:**

The paper introduces a Bayesian variational methodology for trend filtering. Claims in the paper are reasonably well supported by evidence, especially after improvements suggested by the review team.